# A naturally occurring polyacetylene isolated from carrots promotes health and delays signatures of aging

Carolin Thomas [1,7], Reto Erni [2,3,7], Jia Yee Wu [1], Fabian Fischer [1,4], Greta Lamers [1], Giovanna Grigolon[1], Sarah J. Mitchell[5], Kim Zarse [1,6], Erick M. Carreira [2,8] ✉ & Michael Ristow [1,6,8] ✉

To ameliorate or even prevent signatures of aging in ultimately humans, we here report the identification of a previously undescribed polyacetylene contained in the root of carrots (*Daucus carota*), hereafter named isofalcarintriol, which we reveal as potent promoter of longevity in the nematode *C. elegans*. We assign the absolute configuration of the compound as (3 *S*,8 *R*,9 *R*,*E*)-heptadeca-10-en-4,6-diyne-3,8,9-triol, and develop a modular asymmetric synthesis route for all *E*-isofalcarintriol stereoisomers. At the molecular level, isofalcarintriol affects cellular respiration in mammalian cells, *C. elegans*, and mice, and interacts with the α-subunit of the mitochondrial ATP synthase to promote mitochondrial biogenesis. Phenotypically, this also results in decreased mammalian cancer cell growth, as well as improved motility and stress resistance in *C. elegans*, paralleled by reduced protein accumulation in nematodal models of neurodegeneration. In addition, isofalcarintriol supplementation to both wild-type C57BL/6NRj mice on high-fat diet, and aged mice on chow diet results in improved glucose metabolism, increased exercise endurance, and attenuated parameters of frailty at an advanced age. Given these diverse effects on health parameters in both nematodes and mice, isofalcarintriol might become a promising mitohormesis-inducing compound to delay, ameliorate, or prevent aging-associated diseases in humans.

The aging population is an increasing health, social, and financial burden worldwide. For instance, age is a major risk factor for most chronic diseases and strongly increases the prevalence for age-associated diseases including Type-2 diabetes mellitus (T2DM), cancer and neurodegenerative diseases[1]. These pathologies are often caused by common cellular dysregulations, leading to the accumulation of damage with increasing age. Energy metabolism such as insulin

signaling and energy sensing, as well as mitochondrial dysfunction have been identified as hallmarks of aging[2] and might be pharmacologically targeted to ameliorate the age-associated dysfunctions. One of the most effective interventions leading to longevity across species is caloric restriction[3]. Consequently, genetic and pharmacological modulations that mimic energy deficits have been successful in increasing health and lifespan[2]. One crucial mediator within this

[1]Laboratory of Energy Metabolism, Institute of Translational Medicine, Department of Health Sciences and Technology, Swiss Federal Institute (ETH) Zurich, Schorenstrasse 16, 8603 Schwerzenbach, Switzerland. [2]Laboratory of Chemistry and Applied Biosciences, Department of Organic Chemistry, Swiss Federal Institute (ETH) Zurich, Vladimir-Prelog-Weg 1-5/10, Zurich 8093, Switzerland. [3]Biozentrum, University of Basel, Basel 4056, Switzerland. [4]CureVac SE, Tübingen 72076, Germany. [5]Ludwig Princeton Branch, Princeton University, Princeton, NJ 08540, USA. [6]Institute of Experimental Endocrinology, Charité Universitätsmedizin Berlin, Berlin 10117, Germany. [7]These authors contributed equally: Carolin Thomas, Reto Erni. [8]These authors jointly supervised this work: Erick M. Carreira, Michael Ristow. ✉e-mail: carreira@ethz.ch; michael.ristow@charite.de

process is the mitochondrial ATP synthase, as part of the respiratory chain and as main production site of ATP. Inhibition thereof does consequently activate AMP-activated protein kinase (AMPK) signaling, a pathway essential for adaptation to low energy levels. Activation of AMPK is achieved by several health-promoting phytochemical compounds via inhibition of respiratory chain complexes[4,5]. Another key mediator leading to the attenuation of aging is transcription factor nuclear factor erythroid-derived 2-like 2 (NFE2L2, NRF2) which is activated upon oxidative stress and initiates cellular antioxidant responses and adaptations leading to increased oxidative stress resistance. Notably, NRF2 was found to be downregulated in aged organisms[6] and pharmacological activation thereof attenuates the aging phenotype[7]. Various pharmacological approaches have been developed to interfere with these processes, including dietary supplementation with resveratrol, quercetin, epigallocatechin-3-gallate and curcumin[4,5]. However, the bioavailability of these structures is rather poor and some of them showed low stability and controversial clinical efficacy[8-11]. Consequently, there is a strong unmet medical need for the evaluation and identification of novel phytochemicals and naturally occurring anti-aging substances with a potential use as food supplements or drugs to delay the aging process and to prevent the development of associated diseases.

We here performed a screening of 1,200 phytochemical compounds derived from isolated plant extracts for their potential to provoke energy deficit in form of decreased cellular ATP levels. Additionally, we evaluate their ability to activate NRF2 via cellular luciferase reporter system. As a result, a diacetylene derived from the root of *Daucus carota*, hereafter named isofalcarintriol (IFT), shows promising results in both assays. There had been no prior isolation of protocol published to date and, accordingly, the relative and absolute configuration of isofalcarintriol is assigned in this study. Furthermore, we established modular and scalable chemical syntheses for all *E*-isofalcarintriol stereoisomers, enabling the development of molecular probes as well as biological large-scale in vitro and in vivo investigations. From a biological perspective, isofalcarintriol shows health beneficial effects in three different species, including increased lifespan and prolonged motility in models of neurodegeneration in *C. elegans*, decreases cancer cell growth, and improves glucose metabolism, endurance capacity, and parameters of frailty in mice. These effects are most likely caused by direct inhibition of the ATP synthase leading to reactive oxygen species (ROS) signaling, the activation of NRF2 and AMPK, as well as increased mitochondrial mass, i.e. all parameters previously associated with the aging process.

## Results

### A natural polyacetylene acts as an ATP inhibitor and NRF2 activator

As a partial short-term deficit of energy is linked to health and longevity, we have been interested in identifying novel ATP inhibitors. Therefore, we screened a compound library (AnalytiCon Discovery GmbH) of 1,200 single molecules extracted from edible plants derived from Traditional Chinese Medicine (TCM) and herbal treatments for their ability to decrease ATP levels after 15 min of incubation in HepG2 cells. We then analyzed the top candidates for their potential to activate NRF2 and further evaluated effects on life span extension in *C. elegans* (Fig. 1a). In a first step, cellular ATP levels were measured and represented in relative terms as average values of three replicates plus the addition of corresponding standard deviation to this value (Fig. 1b). This allowed us to eliminate false-positive results and focus on molecules that lead to decreased ATP levels. We included three known inhibitors of the mitochondrial ATP synthase, namely piceatannol, Bz-423, and oligomycin as positive controls to define a physiological range of ATP suppression[12]. While oligomycin as a potent inhibitor of the ATP synthase reduced cellular ATP by 20%, piceatannol and Bz-423 do only partially impair ATP generation by 5 to 10% and thereby more

likely act reversibly in a non-toxic manner. Consequently, we decided to evaluate small-molecule candidates displaying inhibition of 5 to 10%, which resulted in 29 hit compounds. Notably, more than half of this list consists of natural small molecules whose structures are unknown (NA) (16 out of 29) (Supplementary Table 1). To narrow down the number of candidates, we performed a second screening for NRF2 activation by using a luciferase-based reporter assay in HEK293 (Fig. 1d). We observed that three out of 29 compounds activated NRF2 after overnight treatment; these included 10-gingerol (structure **2**), alnusone (structure **3**), and a yet undescribed diacetylene (AnalytiCon Discovery GmbH; NP017896) isolated from *Daucus carota*, which we refer to as isofalcarintriol (structure **1**) (Fig. 1c, structure **2** and **3** in Supplementary Fig. 1). Moreover, NRF2 activation was absent in HEK293 cells deficient for *NRF2*, thus validating the specificity of our assay (Fig. 1e).

Remarkably, treatment of HEK293 luciferase reporter cells with isofalcarintriol (**1**) led to a 30-fold increase in NRF2 activation, which exceeded the effects of sulforaphane, an established NRF2 activator[13]. Given the propensity to decrease ATP levels with simultaneous activation of the transcription factor NRF2, we examined the effect on *C. elegans* caused by dietary supplementation with 10-gingerol (**2**), alnusone (**3**) and isofalcarintriol (**1**), respectively. Isofalcarintriol (**1**) had not been previously published, and consequently the proposed structure provided by AnalytiCon needed to be validated and further characterized. Following full characterization, the design and implementation of an asymmetric synthesis route (see **Supplemental Information** for details) ensure sufficient quantities of synthetic isofalcarintriol (structure **1a**) in high purity for further experiments starting from Fig. 1f onwards. Survival assays showed inconsistent lifespan extension by alnusone (**3**) (10 nM) (Supplementary Fig. 1c-d) but not 10-gingerol (**2**) (Supplementary Fig. 1b) in nematodes. Among the three of them, isofalcarintriol (**1a**) represents the most potent substance with a mean life span increase of up to 17% when supplemented with a concentration of 1 nM (Fig. 1f, statistics in Supplementary Table 2 and Supplementary Table 7). Notably, promotion of longevity was also found upon supplementation of 0.1 nM and 10 nM isofalcarintriol, however, to a lesser extend as by 1 nM (**1a**) (Supplementary Fig. 1e-f). Based on these results, we continued evaluating isofalcarintriol (**1a**) in further assays by supplementation of 10 μM to cells and 1 nM and 10 nM to *C. elegans*. Lifespan of *skn-1/NRF2*-deficient nematodes was unchanged upon treatment with isofalcarintriol (**1a**) (Fig. 1g), indicating the essential involvement of NRF2 signaling in the mode of action of this natural polyacetylene.

### Asymmetric synthesis of isofalcarintriol and configurational assignment

To confirm the putative structure of plant-derived isofalcarintriol (**1**) (AnalytiCon Discovery GmbH), we performed nuclear magnetic resonance (NMR) spectroscopy. The spectral data (Supplementary Table 3) were in line with the proposed structure and selected characteristic $^1$H,$^{13}$C-Heteronuclear Multiple Bond Correlation (HMBC) signals are highlighted in Fig. 2a. In $^1$H-NMR, the C-10−C-11 olefin displays vicinal $^3J_{HH}$ couplings constant of 15.4 Hz, supporting the postulated *trans* configuration. Given the lack of information in connection to the relative and absolute configuration of **1**, a synthetic strategy was devised to address the issue. Retrosynthetically (Fig. 2b) we envisioned a modular approach for **1** wherein alkyne **4a** or **4b** were joined with bromoalkyne **5** at late-stage by Cadiot−Chodkiewicz cross-coupling reaction; detailed analytical data are contained in the **Supplementary Information**. The syntheses of all four syn-1,2-diol containing isofalcarintriol (**1a,b**) stereoisomers are depicted in Fig. 2c. In summary, a modular asymmetric synthesis was developed giving access to all eight *E*-isofalcarintriol (**1**) stereoisomers. To determine the absolute configuration of the 3-OH and optical purity, an enantioselective supercritical fluid chromatography (eSFC) separation method

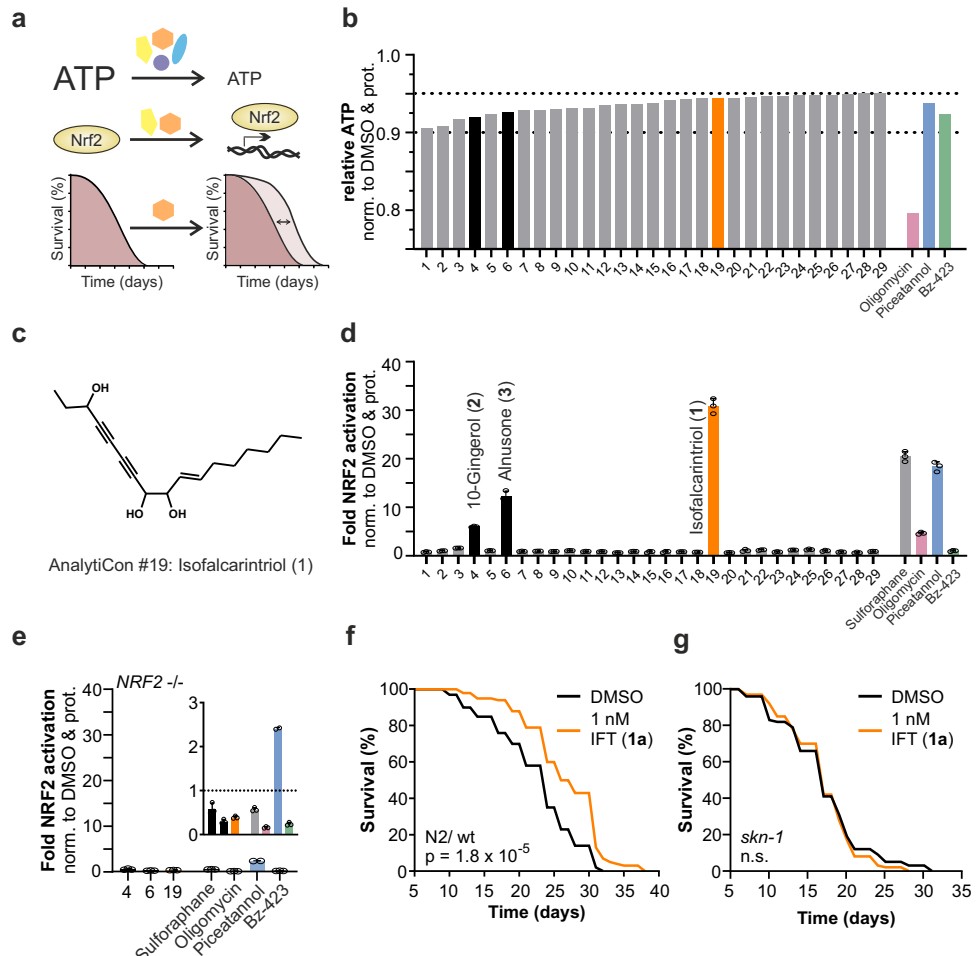

**Fig. 1 | Identification of a natural structure, isofalcarintriol (IFT) as potent suppressor of ATP generation and activator of NRF2. a** Schematic representation of the compound screening procedure, including an initial ATP screening and NRF2 activation assay in cells, followed by lifespan assays in *C. elegans*, **b** ATP assay screening revealed 29 small-molecule candidates (5 µg/ ml) that inhibit cellular ATP levels by 5–10% (average plus SD) compared to DMSO solvent control after 15 min of incubation. 10 µM of oligomycin, piceatannol, and Bz-423 were used as positive controls. **c** Structure of compound AnalytiCon Discovery GmbH #19 isofalcarintriol (**1**) as provided by AnalytiCon Discovery GmbH after extraction of *Daucus carota*. **d** NRF2 luciferase reporter assay with top ATP inhibitors after overnight treatment in transgenic HEK293 cells. 10-gingerol (**2**), alnusone (**3**), and isofalcarintriol (**1**)

were identified as potent NRF2 activators. Sulforaphane was used as positive control. **e** NRF2 activation over solvent control in a HEK293 homozygote *NRF2* deletion reporter cell line, including #19 (**1**). **f** Lifespan of WT *C. elegans* (N2) upon treatment with (synthetic) isofalcarintriol (**1a**) (1 nM). **g** Lifespan of *C. elegans* when supplementing isofalcarintriol (**1a**) to *skn-1* (*NRF2*) deficient nematodes. Cell data include three technical replicates and are represented as the sum of average + SD, or average + SD. *C. elegans* data include three biologically independent samples and are represented as average. Statistics: log-rank test, one-way ANOVA and Dunnett's or Bartlett's posthoc test. $p < 0.0001 = ****$. Source data are provided as a Source Data file.

was developed (Supplementary Fig. 3a-e). Combined, these observations lead to the assignment of the absolute configuration of naturally occurring isofalcarintriol (**1**) and establish an enantiomeric excess of the authentic sample at >95%. Gram-scale synthesis of (3 *S*,8 *R*,9 *R*)-isofalcarintriol (**1a**) was achieved, demonstrating the robustness and scalability of the route. NRF2 reporter assay shows peak activation with 10 µM at 26-fold activation by **1a** and 25-fold activation by **1b**, respectively, whereas *ent*-**1a** and *ent*-**1b** were inactive (Fig. 2d). Thereby reconfirming absolute assignment and showcasing the key importance of 3-hydroxyl configuration for NRF2 activation. Assessment of the corresponding *anti*-1,2-diols (**1c,d, ent-1c,d**) confirm this observation (Supplementary Fig. 2b). To the best of our knowledge, no isolation protocol of isofalcarintriol (**1a**) has been reported to date. To quantify the natural abundance of isofalcarintriol (**1a**) and its spatial distribution in *D. carota* roots an extraction procedure and liquid chromatography–mass spectrometry (LC-MS) separation method were developed (**Supplementary Information, Chemical Synthesis Procedures**). The natural abundance of isofalcarintriol (**1a**) was

estimated at 3.8–8.9 µg/g of dry weight (assuming 90% water content) depending on the extraction solvent used (Supplementary Table 4).

## Inhibition of the mitochondrial ATP synthase and AMPK signaling

We assessed potent cellular interaction partners of isofalcarintriol (**1a**) by performing quantitative proteomics. Biotin-isofalcarintriol (structure **15**, for synthesis see Supplementary Figs. 5, 6) was used in two independent pulldown assays with mass spectrometric analysis in HepG2 and HEK293 cells, respectively. Due to the absence of biotin-transporters in HEK293 cells, treatment with biotin-isofalcarintriol (**15**) was conducted in intact HepG2 cells, whereas HEK293 were lysed before incubation with the compound thus circumvent impaired cellular uptake of biotin (Supplementary Fig. 7). Three independent replicates were processed in parallel and pooled before mass spectrometric measurement. The analysis was performed using a 1.5-fold increase in intensity over biotin-alkene (**S12**) (negative control), as well as a minimum detection of 2 peptides per protein as lower cut off.

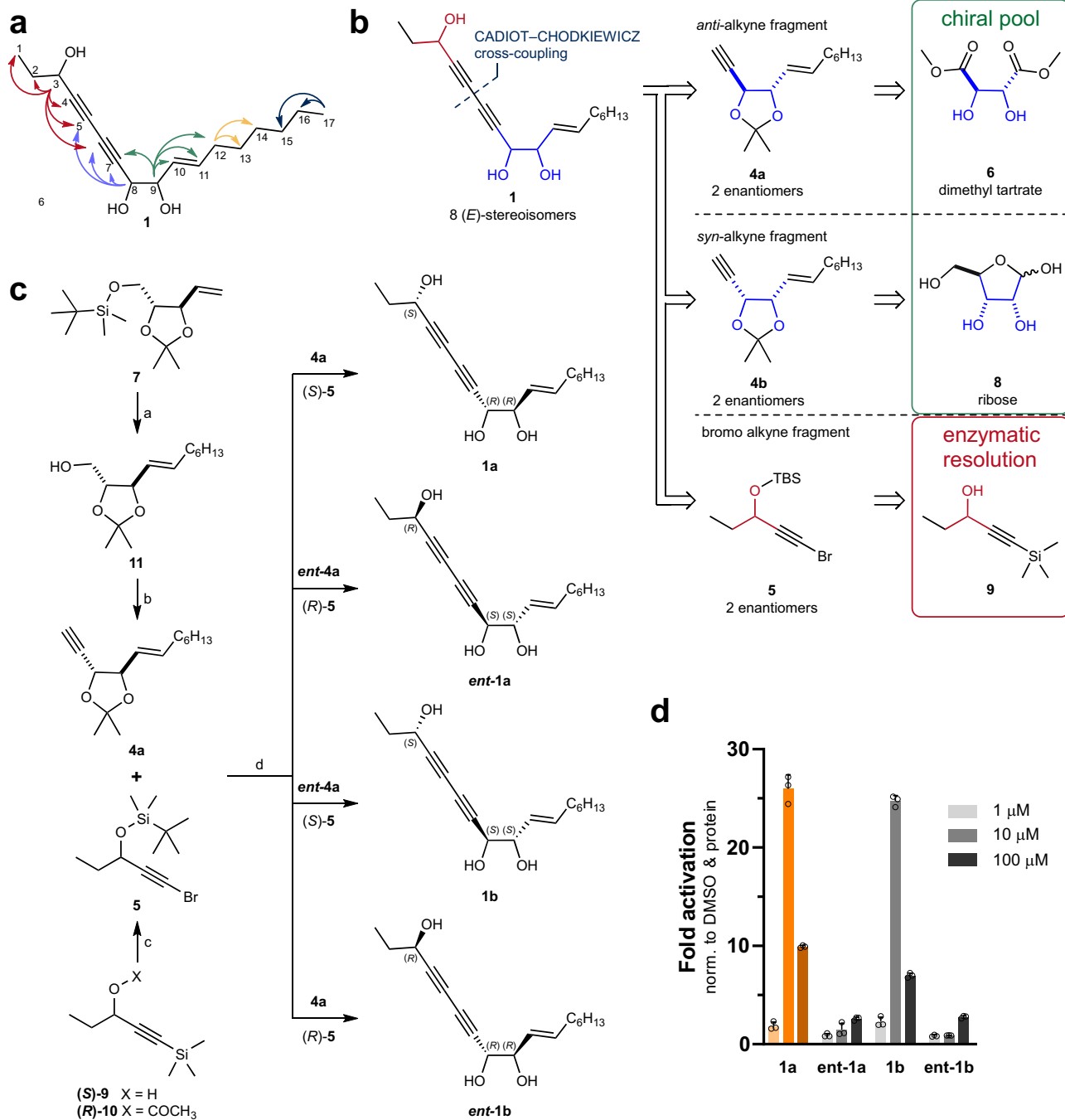

**Fig. 2 | Structure elucidation of isofalcarintriol by asymmetric synthesis.**
**a** Selected $^1$H,$^{13}$C-HMBC correlations observed in isofalcarintriol (**1**).
**b** Retrosynthetic analysis of isofalcarintriol (**1**) tracing back to chiral pool starting materials **6**, **8** and previously reported enantiopure propargylic alcohol **9**.
**c** Synthesis scheme of all four *syn*-1,2-diol containing isofalcarintriol (**1a**, **b**) stereoisomers. Reagents and conditions: a, 1: 1-octene, Grubbs Catalyst 2nd Generation catalytic (cat.), CH$_2$Cl$_2$, 40 °C; 2: tetra-*n*-butylammonium fluoride, THF, 0 °C to RT, 64–71% over two steps. b, 1: Dess–Martin periodinane (DMP), CH$_2$Cl$_2$, RT; 2: Ohira–Bestmann reagent, K$_2$CO$_3$, MeOH, 0 °C, 72–76% over 2 steps. c, 1: K$_2$CO$_3$,

MeOH, 40 °C; 2: *t*-butyldimethylsilyl chloride (TBSCl), imidazole, CH$_2$Cl$_2$, 0 °C to RT; 3: *N*-bromosuccinimide, AgNO$_3$ cat., acetone, RT, 38–59% over three steps. d, 1: **4a**, CuCl cat., *n*-BuNH$_2$ (aq.), Et$_2$O, RT, then **5**, 0 °C to RT, 61–91%; 2: CF$_3$COOH, THF/water (4:1), 40 °C or HCl (aq.), MeOH, RT, 87–99%. **d** NRF2 luciferase reporter assay after overnight treatment in transgenic HEK293 cells where only (3 *S*,8 *R*,9 *R*)-isofalcarintriol (**1a**) and (3 *S*,8 *S*,9 *S*)-isofalcarintriol (**1b**) activated NRF2, underlining the importance of the configuration of the 3-hydroxy group on activity. Data include three technical replicates and are represented as average + SD. Source data are provided as a Source Data file.

Enriched proteins from both experiments were combined and the ATP5A (α-subunit) and ATP5O (OSCP) as part of the mitochondrial ATP synthase were identified as potential common interaction partners of isofalcarintriol (Fig. 3a, Supplementary Table 5). Interestingly, a functional annotation analysis revealed that the category "Formation of ATP by chemiosmotic coupling", including both subunits of the ATP

synthase, was the most highly overrepresented (53x) biological function compared to other significantly enriched cellular pathways such as cellular transport processes and signaling by GTPases (Supplementary Table 6). Together with the data on partial ATP depletion the evidence points towards mitochondrial energy metabolism being the main pathway targeted by IFT. In addition, impairment of either

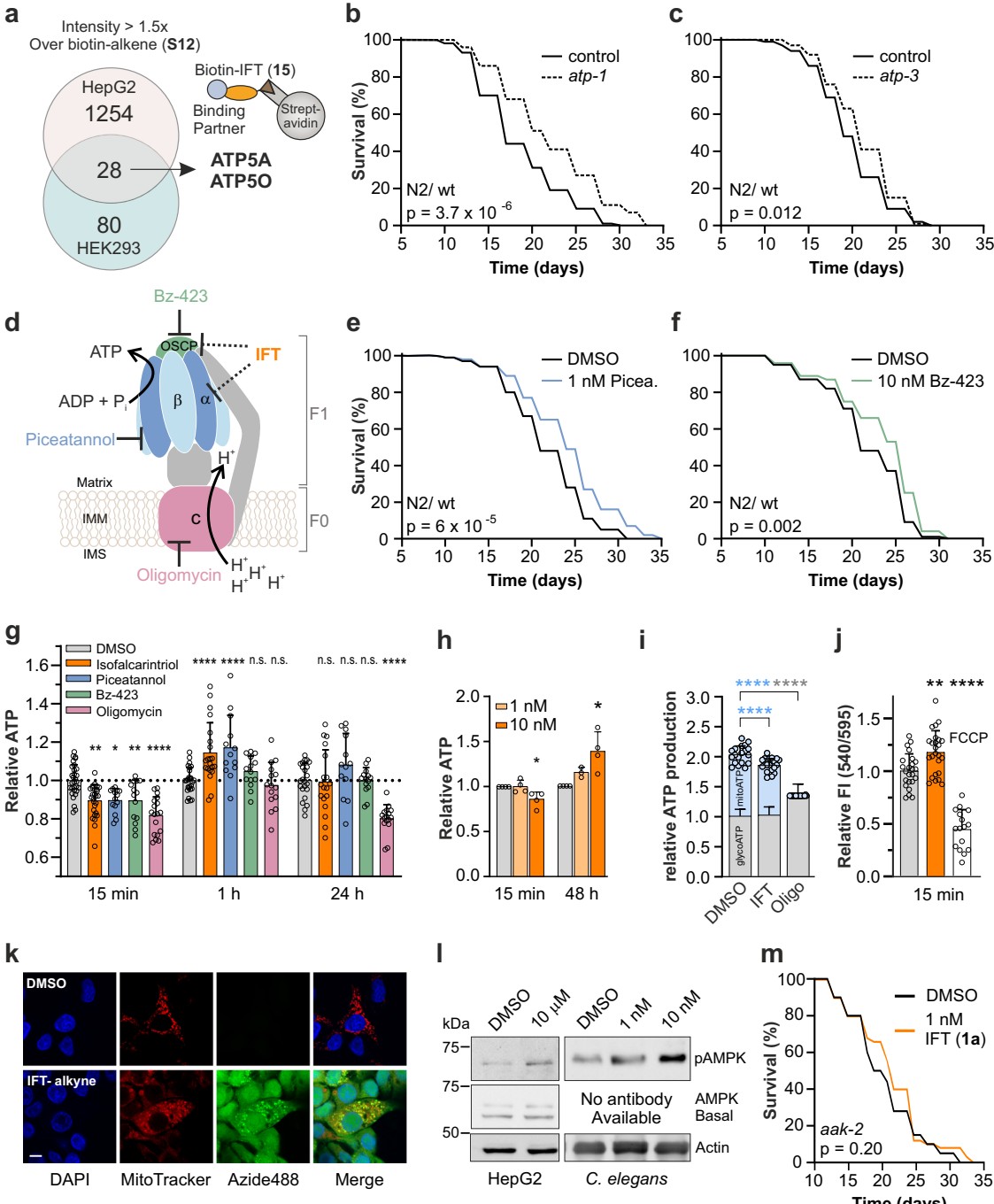

**Fig. 3 | Isofalcarintriol (IFT) potentially interacts with the mitochondrial ATP synthase, resulting in impaired mitochondrial ATP production and activation of AMPK. a** Biotin pulldown with HepG2 and HEK293 cells and mass spectrometric analysis identifying subunits ATP5A (α-SU) and ATP5O (OSCP) as interaction partners of biotin-isofalcarintriol (**15**). **b, c** Lifespan upon RNAi-based downregulation of *C. elegans* (**b**) α-SU = *atp-1* and (**c**) OSCP = *atp-3*. **d** Schematic representation of the ATP synthase. IMM: Inner mitochondrial membrane; IMS: Intermembrane space. Pharmacological inhibition by (**e**) piceatannol (α-SU) and (**f**) bz-423 (OSCP) in *C. elegans*. **g** Time-response of ATP dynamics upon isofalcarintriol (**1a**) treatment in HepG2 at 15 min (DMSO: *n* = 30; IFT: *n* = 27) (*p* = 0.032), 1 h (DMSO: *n* = 27; IFT: *n* = 21) (*p* = <0.0001) and 24 h (DMSO: *n* = 23; IFT: *n* = 18). **h** Time-response of ATP dynamics upon isofalcarintriol (1a) treatment in *C. elegans* at 15 min (*p* = 0.045) and 48 h (*p* = 0.022) (15 min: *n* = 4 per condition; 48 h DMSO: *n* = 3; 48 h IFT: *n* = 4). **i** Real-Time ATP Rate Assay quantifying mitochondrial (*p* = <0.0001) and glycolytic

ATP production upon injection of isofalcarintriol (1a) (*n* = 20 independent cell samples per condition). **j** Mitochondrial membrane potential upon isofalcarintriol (1a) treatment (DMSO: *n* = 24; IFT: *n* = 25) (*p* = 0.003). **k** Isofalcarintriol-alkyne localization by immunostaining of HepG2. Fluorophore-labeling of isofalcarintriol-alkyne was achieved via click chemistry using azide488, while DMSO + azide488 control treated cells did not show any signal. Scale bar: 10 μM. **l** Western blot of cell and *C. elegans* samples indicate isofalcarintriol (1a)-dependent AMPK activation by phosphorylation. Uncropped blot in Source Data. **m** Lifespan of Isofalcarintriol (1a)-treated lifespan in *aak-2* (*AMPK*)-deficient nematodes. Data are represented as average or average + SD. **g, h, j, k** Measured in ≥ 3 independent biological experiments, respectively. Statistics: log-rank test, one-way ANOVA & Barlett's Posthoc test, two-way ANOVA & Dunnett's or Tukey Posthoc test. Source data are provided as a Source Data file.

*atp-1*/ATP5A or *atp-3*/ATP5O by RNAi feeding resulted in an increased life expectancy of *C. elegans*, emphasizing that partial impairment of the ATP synthase is lifespan extending (Fig. 3b). Moreover, epistasis with simultaneous isofalcarintriol (**1a**) treatment and *atp-1* down-regulation results in significantly decreased *C. elegans* life expectancy compared to *atp-1* RNAi treatment alone, which further validates a strong dependency of isofalcarintriol (**1a**) on ATP synthase functionality (Supplementary Fig. 8a). Similar to genetic impairment, pharmacological inhibition of the F1 ATP synthase by piceatannol and the inhibition of the peripheral stalk by Bz-423 (Fig. 3d) induces a beneficial effect on lifespan (Fig. 3e, f). However, inhibition of the c subunit of the ATP synthase by oligomycin does not affect lifespan (Supplementary Fig. 8c).

We furthermore evaluated the time-dependent ATP dynamics after treatment with isofalcarintriol (**1a**) (Fig. 3g). Notably, isofalcarintriol (**1a**) causes a decrease of cellular ATP levels after 15 min and leads to a transient increase after 1 h before normalization back to baseline. While known reversible F1 ATP synthase inhibitors showed similar ATP dynamics, ATP production could not be recovered after 24 hours of oligomycin treatment. These results indicate a reversible mode of energy inhibition by isofalcarintriol (**1a**), similar to piceatannol and Bz-423. In accordance with these cell culture data, *C. elegans* ATP levels were significantly decreased after short-term supplementation with 10 nM isofalcarintriol (**1a**) (Fig. 3h). Surprisingly and in contrast to observed effects in cells, ATP levels in *C. elegans* stayed elevated long-term. Fitting to our hypothesis of an isofalcarintriol-dependent impairment of the mitochondrial ATP synthase, we could demonstrate a deficiency of mitochondrial ATP production after injection of the compound while the glycolytic ATP production rate was unaffected (Fig. 3i, Supplementary Fig. 8c, d). In addition, we detected a significant increase in membrane potential after 15 minutes (Fig. 3j) which results from the loss of proton flux across the inner mitochondrial membrane, as typically observed with other ATP synthase inhibitors. No changes in membrane potential were seen to later time points indicating a transient nature of this effect (Supplementary Fig. 8e).

To confirm the presence of isofalcarintriol (**1a**) within mitochondria and validate its compartmental proximity to the ATP synthase, we performed immunofluorescent stainings. To visualize isofalcarintriol (**1a**), we attached a fluorophore to isofalcarintriol-alkyne (structure 3 *S*,8 *R*,9*R*-**27**, for synthesis and NRF2 activations see Supplementary Fig. 9a, b) via copper-catalyzed click chemistry post-treatment. Besides being present in diverse other subcellular localizations, we could prove its accumulation in mitochondria by co-localization with the Mito-Tracker signal (Fig. 3k). During confocal microscopy, we observed potential indications of differences in shape between round isofalcarintriol-treated mitochondria and DMSO-treated tubular-shaped mitochondria (Supplementary Fig. 8f). These observations are in accordance with Fu and Lippincott-Schwartz[14], who reported an increase of round-shaped mitochondria due to elevated mitochondrial fission as consequence of oligomycin-dependent ATP synthase inhibition. However, to further quantify differences in mitochondrial dynamics, advanced microscopy analyses would need to be conducted in future studies. As a consequence of an IFT-induced initial energy deficit, downstream AMPK signaling is activated by phosphorylation within 1 h after treatment with isofalcarintriol (**1a**) in cells and *C. elegans* as shown via immunoblotting (Fig. 3l). AMPK signaling appears to play an indispensable role in isofalcarintriol-dependent lifespan extension as *aak-2*/AMPK-deficient nematodes supplemented with isofalcarintriol (**1a**) are not long-lived (Fig. 3m).

To further validate a mitochondria-targeted mode of action, parameters of respiration were assessed upon isofalcarintriol (**1a**) treatment in Seahorse XF Cell Mito Stress Assays. Injection of 10 μM isofalcarintriol (**1a**) to HepG2 cells (Fig. 4a) provoked an initial decrease in oxygen consumption rate (OCR) (Fig. 4b) and a decrease

in ATP production (Fig. 4c) without affecting maximal respiration after FCCP addition (Fig. 4d). Similar to this, overnight treatment with isofalcarintriol (**1a**) (Fig. 4e) caused a decrease of basal respiration (Fig. 4f), ATP production (Fig. 4g), and maximal respiration (Fig. 4h) compared to control cells. Notably, the degree of partial OCR inhibition after short- and long-term treatment with isofalcarintriol (**1a**) was observed in a similar magnitude as piceatannol-dependent impairment of respiration, but was more effective than application of Bz-423. In contrast, oligomycin as a strong inhibitor of the c-subunit of ATP synthase blocks mitochondrial respiration almost completely. Isofalcarintriol-dependent effects on respiration could be confirmed in *C. elegans* (Fig. 4i) by a significantly decreased basal OCR after overnight treatment with 10 nM isofalcarintriol (**1a**) (Fig. 4f), while maximal respiration was unaffected upon isofalcarintriol (**1a**) supplementation (Fig. 4k). Interestingly, oxygen consumption (active phase and whole day) (Fig. 4l, Supplementary Fig. 10a, f) and carbon dioxide production (active phase) (Supplementary Fig. 10b, f) was also attenuated in young female wild-type mice on chow diet, as evaluated via indirect calorimetry. A similar trend ($p = 0.09$ for OCR and $p = 0.06$ for carbon dioxide production) was seen in male mice (Supplementary Fig. 10c, d, e).

## Oxidative stress resistance via mitohormetic ROS signaling

To further investigate molecular mechanisms and the impact of isofalcarintriol (**1a**) on health parameters, we performed a set of biochemical assays in cells and *C. elegans*. In accordance with its NRF2-activating properties (Fig. 1), we saw an initial isofalcarintriol-mediated ROS signal in cells (Fig. 5a) and C. *elegans* (Fig. 5b) after 15 minutes of treatment. However, long-term ROS levels were significantly decreased in both cells and *C. elegans*, possibly via adaptation processes through downstream effectors such as NRF2, as previously observed for other compounds[15] and biological settings[16]. To confirm the role of ROS in isofalcarintriol-mediated NRF2 activation, a NRF2 luciferase assay was performed in reporter cells overexpressing human catalase in the cytoplasm (Fig. 5c). Catalases are major enzymatic antioxidants crucially involved in detoxification of hydrogen peroxide and thereby suppressing the formation of reactive oxygen species and oxidative stress. Indeed, NRF2 activation by isofalcarintriol (**1a**) was significantly decreased when catalase was overexpressed compared to empty vector control cells, indicating the need of an adequate ROS signal as upstream element of isofalcarintriol-dependent regulation of ROS homeostasis. In an analogous experiment, overexpression of cytosolic catalase 1 (*ctl-1*) in *C. elegans* suppressed the lifespan extending effect of isofalcarintriol (**1a**) and once more emphasizes the importance of ROS signaling (Fig. 5d). On top of that, isofalcarintriol (**1a**) appears to increase oxidative stress resistance and prolongs survival of *C. elegans* in a paraquat-induced oxidative stress assay (Fig. 5e). These observations resemble the process of *mitohormesis*, as previously described[17–19].

## Attenuation of age-related pathologies

In addition to the modulation of cellular signaling pathways, isofalcarintriol (**1a**) treatment shows profound anti-aging properties in several models of age-associated dysregulation. By using simplified *C. elegans* models of neurodegeneration, we could resemble basic protein accumulation phenotypes of Huntington's disease and Alzheimer's disease, leading to movement impairments and complete paralysis, respectively. Several hours after induction, overexpression and consequent aggregation of human Aβ caused total paralysis of the nematodes in an Alzheimer's disease model (Fig. 5f). Interestingly, the onset and overall development of paralysis was strongly delayed upon 1 nM isofalcarintriol (**1a**) treatment, representing a health beneficial effect of isofalcarintriol (**1a**) against pathological protein accumulation in neurodegenerative diseases. To evaluate Huntington-related protein accumulation, we used two different nematode strains, either

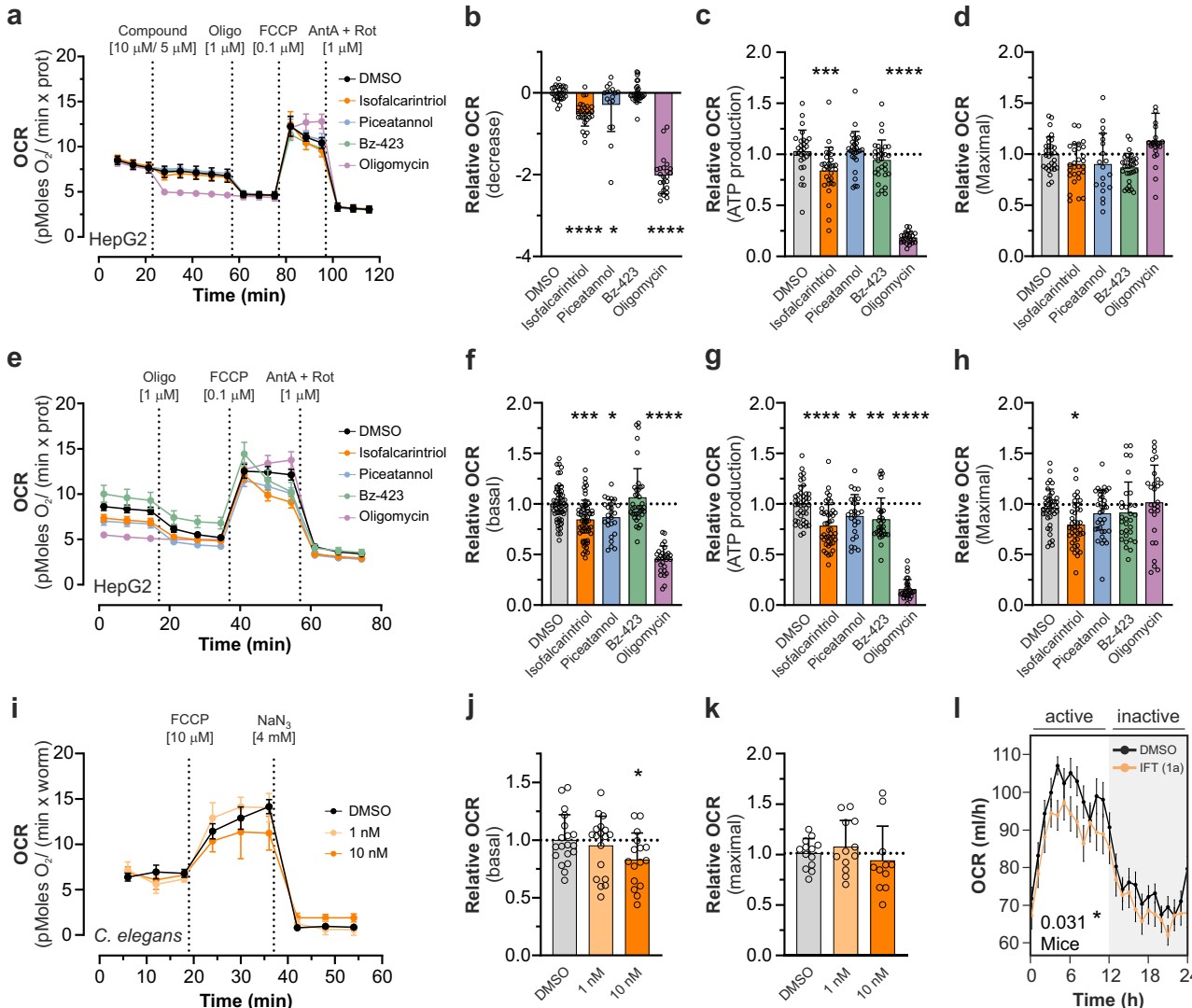

**Fig. 4 | Isofalcarintriol (IFT) impairs mitochondrial respiration in cells, *C. elegans* and mice. a** Seahorse XF Cell Mito Stress Assay showing the OCR development over time after injection of isofalcarintriol (**1a**) (10 μM) and positive controls in HepG2 cells (DMSO $n = 12$; IFT $n = 11$ independent cell samples per condition). Quantification of respiration parameters in a, including (**b**) a decrease of respiration (IFT: $p < 0.0001$) (**c**) ATP production (calculated) (IFT: $p = 0.001$) and (**d**) maximal respiration. **e** Seahorse XF Cell Mito Stress Assay showing the OCR development over time after overnight pre-treatment with isofalcarintriol (**1a**) (10 μM) and positive controls in HepG2 cells. $n = 12$ independent cell samples per condition. Quantification of respiration parameters in (**e**), including (**f**) basal respiration (IFT: $p = 0.0003$), (**g**) ATP production (calculated) (IFT: $p < 0.0001$) and (**h**) maximum

respiration (IFT: $p = 0.016$). **i** Seahorse XF Cell Mito Stress Assay showing the OCR development over time after overnight pre-treatment of isofalcarintriol (**1a**) (1 nM, 10 nM) in *C. elegans* (DMSO and IFT 1 nM: $n = 7$; IFT 10 nM: $n = 5$ biologically independent samples per condition). Quantification of respiration parameters, including (**j**) basal respiration (IFT 10 nM: $p = 0.038$) and (**k**) maximal respiration. **l** Indirect calorimetry of female C57BL/6NRj mice (DMSO: $n = 10$; IFT: $n = 11$ mice) on chow diet, and quantification of OCR upon treatment with isofalcarintriol (**1a**) (0.1 mg/kg body weight) ($p = 0.031$). Data are represented as average ± SD, + SD or ± SEM (**l**), and (**b**)–(**d**), (**f**)–(**h**), and (**j**)–(**k**) were measured in ≥ 3 independent experiments. Statistics: one-way ANOVA with or without Dunnett's or Bonferroni's post-hoc test. Source data are provided as a Source Data file.

expressing 19 (AM23; healthy control) or 67 (AM716; pathological model) glutamine repeats within the huntingtin protein. Due to immense protein accumulation, AM716 nematodes showed impaired movement which was quantified by a thrashing assay. Thereby, 1 nM isofalcarintriol (**1a**) treatment partially rescued the movement deficiency of AM716 nematodes, indicating a potential beneficial role in Huntington's disease (Fig. 5g).

Furthermore, we analyzed tumor cell growth, as the occurrence of cancer strongly correlates with age. An initial dose-response experiment revealed a specific growth inhibition of breast cancer cells (MCF-7) by isofalcarintriol (**1a**) compared to non-tumor human mammary epithelial cells (HMEpC) that were significantly less affected by this treatment (Fig. 5h, Supplementary Fig. 11b). Interestingly, isofalcarintriol (**1a**) holds a dual role in inhibiting MCF-7 growth while

boosting HMEpC proliferation in low to moderate compound concentrations between 0.1 μM and 1 μM. This might indicate a context-based mode of action by isofalcarintriol (**1a**). In addition, isofalcarintriol (**1a**) impairs proliferation of other tumor cell lines including HepG2 (human liver cancer) and HT-29 (human colon cancer) when applied in concentrations above 1 μM (Supplementary Fig. 11a, b). To evaluate its effect on the metastatic potential of a cell line, we performed soft agar assays with MCF-7, HepG2, and HT-29 cells (Fig. 5i). We could demonstrate a striking effect of isofalcarintriol (**1a**) on the ability to form cell colonies in a non-solid surrounding. While metastatic tumor cells can easily grow under these conditions, isofalcarintriol-treated cells lack this tumorigenic property widely, which manifests in an almost complete extinction of HepG2 and HT-29 colonies, as well as a partial reduction of number of colonies

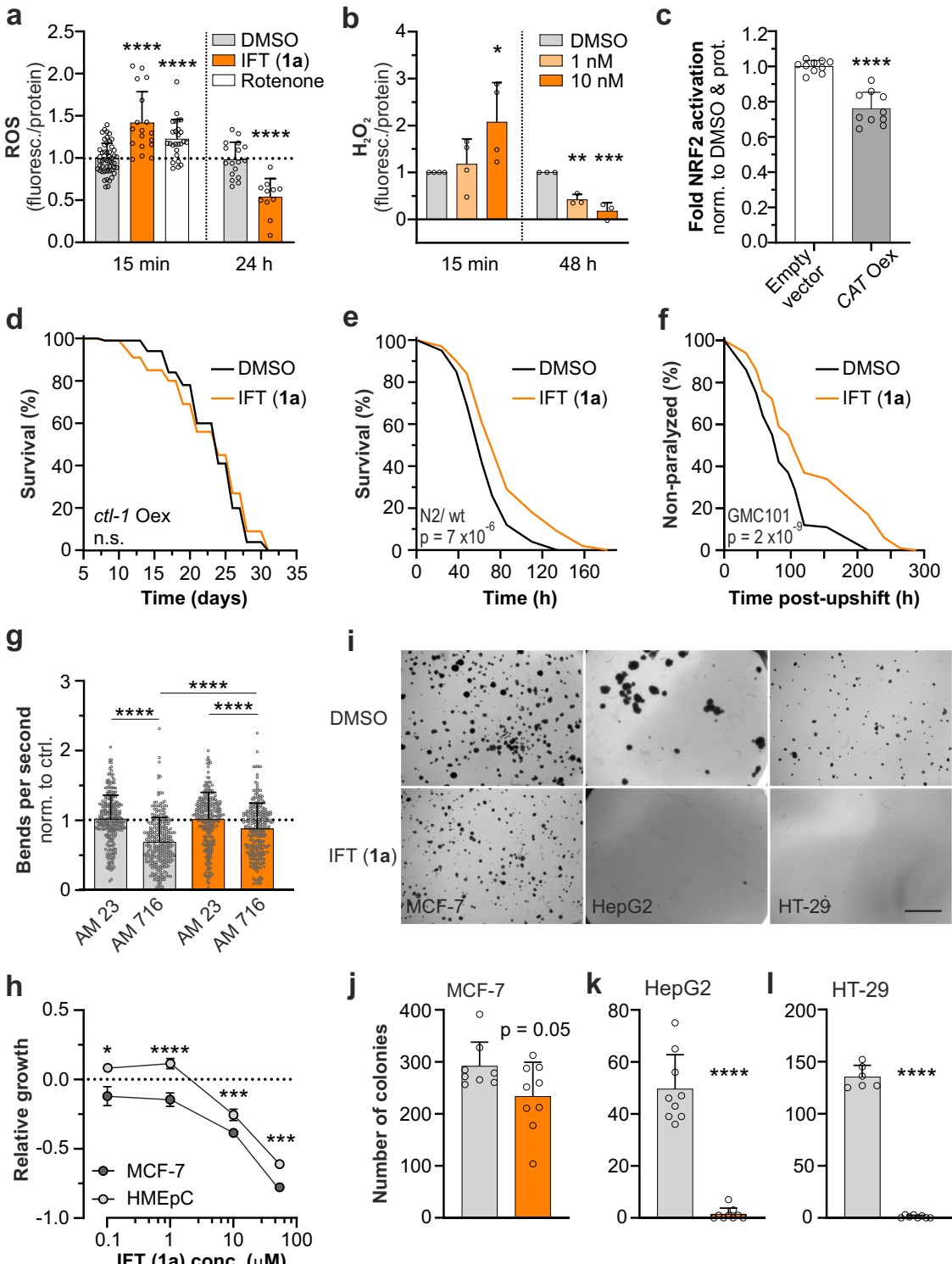

(Fig. 5j, k, l) and a diminished colony size of MCF-7 cells (Supplementary Fig. 11c).

## Metabolic phenotypes and exercise capacity of obese mice

Next, we investigated the translational potential of isofalcarintriol (**1a**) by its supplementation to the drinking water of wild-type C57BL/6NRj mice on high-fat diet (0.1 mg/kg body weight) (Fig. 6a). Most importantly, a prior two-week toxicity study in animals on chow diet did not show any elevation of liver toxicity parameter such as ALAT and ASAT levels in the blood when applying a maximum dose of 2.5 mg/kg

isofalcarintriol (Supplementary Fig. 12a–d), assuming that long-term treatment with the compound in lower doses is save as well. The high-fat diet applied in this study serves to induce overweight and challenge the metabolic phenotype, provoking a decrease in glucose sensitivity and promoting insulin resistance as typically seen in obese and diabetic individuals. During the 6 months course of this high-fat diet study, no major differences in body mass (Supplementary Fig. 13a), body composition (fat mass and lean mass) (Supplementary Fig. 13b, c), blood lipid levels (Supplementary Fig. 13d–k), and liver toxicity parameters (Supplementary Fig. 13l–o) were detected between treated

**Fig. 5 | Isofalcarintriol promotes oxidative stress resistance and targets age-related pathologies. a** DCF-DA and (**b**) AmplexRed determination of ROS dynamics in HepG2 (15 min DMSO $n = 50$; 15 min IFT: $n = 18$; 24 h DMSO: $n = 17$; 24 h IFT: $n = 11$ independent cell samples per condition) (15 min: $p < 0.0001$); 24 h: $p < 0.0001$) and *C. elegans* (15 min: $n = 4$; 48 h: $n = 3$ biologically independent samples per condition) (15 min: $p = 0.018$; 48 h: $p = 0.002$ for 1 nM and $p = 0.0003$ for 10 nM), respectively. **c** NRF2 luciferase reporter activation in cells overexpressing human catalase (*CAT* Oex) upon isofalcarintriol (**1a**) treatment (10 μM) compared to empty vector control ($n = 10$ independent cell samples per condition) ($p < 0.0001$). **d** Lifespan of *ctl-1* (cytosolic catalase)-overexpressing nematodes. **e** Paraquat stress assay (10 mM paraquat) of *C. elegans* treated with isofalcarintriol (**1a**) (1 nM) compared to DMSO control. **f** Paralysis assay with GMC101 nematodes, a protein aggregation model of Alzheimer's disease upon Isofalcarintriol (**1a**) treatment. **g** Motility assay with AM23 (control strain) and AM176 (CAG repeat strain) nematodes, as a protein aggregation model of Huntington's disease (AM23 DMSO:

$n = 277$; AM23 IFT: $n = 258$; AM716 DMSO: $n = 276$; AM716 IFT: $n = 277$ nematodes per condition in two independent experiments, $p < 0.0001$ for each comparison). **h** Proliferation assay with MCF-7 and its non-tumor control cells, HMEpC (MCF-7 0.1 and 50 μM: $n = 6$; MCF-7 1 and 10 μM: $n = 3$; HMEpC: $n = 3$ independent cell samples per condition) after 96 h treatment with of isofalcarintriol (**1a**) (0.1 μM: $p = 0.01$; 1 μM: $p < 0.0001$, 10 μM: $p = 0.0006$, 50 μM: $p = 0.0005$). **i** Soft agar colony formation assay. **j** MCF-7 ($p = 0.05$) (DMSO: $n = 8$; IFT: $n = 9$ independent cell samples per condition), **k** HepG2 ($p < 0.0001$) ($n = 9$ independent cell samples per condition), and **l** HT-29 ($p < 0.0001$) when treated with isofalcarintriol (**1a**) (DMSO: $n = 6$; IFT: $n = 8$ independent cell samples per condition). Scale bar: 1 mm. Data are represented as average or average ± SD or + SD. Data shown in (**a**), (**b**), (**h**), (**i**), and (**j**) were measured in ≥ 3 independent experiments. Statistics: log-rank test, two-sided unpaired student's *t*-test, one-way ANOVA and Dunnett's posthoc test, two-way ANOVA and Sidak's post-hoc test. Source data are provided as a Source Data file.

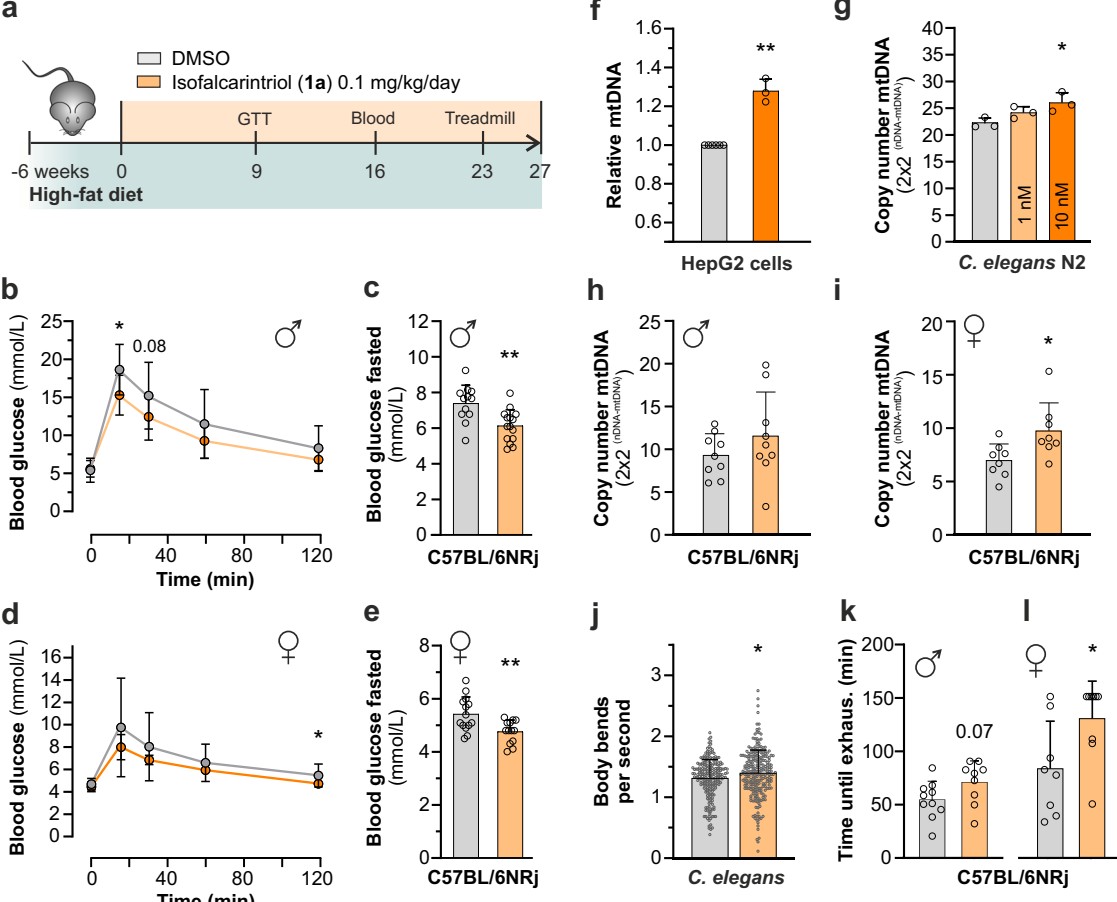

**Fig. 6 | Isofalcarintriol improves glucose metabolism in obese mice and acts as exercise mimetic though increased mitochondrial biogenesis. a** Schematic overview of isofalcarintriol (**1a**) supplementation study with C57BL/6NRj mice on high-fat diet. **b** Glucose tolerance test with male mice (DMSO: $n = 12$ mice; IFT: $n = 15$ mice) (15 min: $p = 0.01$). **c** Fasted blood glucose level in male mice (DMSO: $n = 12$ mice; IFT: $n = 15$ mice) upon isofalcarintriol (**1a**) supplementation after 16 weeks ($p = 0.0022$). **d** Glucose tolerance test with female mice ($n = 14$ mice per treatment) after 9 weeks of treatment (120 min: $p = 0.02$). **e** Fasted blood glucose level in female mice (DMSO: $n = 14$ mice; IFT: $n = 13$ mice) upon isofalcarintriol (**1a**) supplementation ($p = 0.0048$). **f–i** Mitochondrial mass (mtDNA/nDNA ratio) after supplementation of isofalcarintriol (**1a**) in (**f**) HepG2 cells (DMSO: $n = 6$; IFT: $n = 3$

independent cell samples) ($p = 0.0014$). **g** *C. elegans* ($n = 3$ biologically independent samples per condition) (10 nM: $p = 0.0238$), and (**h**) male ($n = 9$ mice per treatment) and (**i**) female ($n = 8$ mice per treatment) high-fat diet mouse muscle tissue (*M. gastrocnemius*) ($p = 0.0222$). **j** Thrashing assay quantifying motility of *C. elegans* on day 5 of adulthood (DMSO: $n = 221$; IFT = 248 nematodes in two independent experiments) ($p = 0.0134$). Treadmill exercise with C57BL/6NRj (**k**) male (DMSO: $n = 10$; IFT: $n = 9$ mice) ($p = 0.07$) and (**l**) female mice (DMSO: $n = 8$; IFT: $n = 9$ mice) ($p = 0.0318$) on high-fat diet. All data are represented as average + SD or ± SD. Statistics: two-sided unpaired student's *t*-test; two-way repeated measures ANOVA; one-way ANOVA and Dunnett's post-hoc test. Source data are provided as a Source Data file.

and control animals. After 9 weeks of treatment, we performed a glucose tolerance test (GTT) and observed an increased glucose sensitivity of isofalcarintriol-treated animals compared to control animals in both sexes (Fig. 6b, d). To this time point, basal fasted blood glucose

levels were unchanged. However, from week 16 onwards, fasted blood glucose levels were significantly reduced in both, males and females (Fig. 6c, e), indicating a positive effect of isofalcarintriol (**1a**) treatment on the phenotype of diabetes. Besides that, isofalcarintriol treatment

mediated a rise in mitochondrial mass by 27%, 17%, and 26% in mammalian cells (Fig. 6f), *C.elegans* (Fig. 6g), and murine muscle tissue (Fig. 6h, i), respectively. In line with this enhanced mitochondrial biogenesis, we also observed increased motility of *C. elegans*, manifesting in higher frequency of body bends during a thrashing assay on day 5 of treatment (Fig. 6j). Most surprisingly, we also quantified a significant improvement in treadmill endurance capacity in our high-fat diet cohort (Fig. 6k, l).

## Frailty and health parameters in aged mice

To have a direct read-out of isofalcarintriol-mediated anti-aging effects, we started to supplement the compound to aged wild-type C57BL/6NRj mice on chow diet at the age of 16 months and assessed metabolic measures as well parameters of frailty until the natural death of the animals (Fig. 7a). Isofalcarintriol (**1a**) treatment did not affect body mass (Supplementary Fig. 14a, c), body composition (Supplementary Fig. 14b, d), or blood lipid levels (Supplementary Fig. 14e-l) compared to control animals. Similar to high-fat diet animals, aged male mice on isofalcarintriol (**1a**) treatment had a significantly increased glucose sensitivity (Fig. 7b), while basal fasted blood glucose levels were unchanged (Fig. 7c). In contrast, aged female mice did not differ in glucose sensitivity during a glucose tolerance test (Fig. 7d) but showed a tendency for decreased fasted blood glucose levels (Fig. 7e).

Although we did not see an increase in murine lifespan (Supplementary Fig. 15a, b) we have strong evidence that isofalcarintriol prolongs the health span of aged mice. For example, we assessed the frailty index score[20,21], consisting of the evaluation of 31 single parameters previously associated with the aging process. Notably, isofalcarintriol treated mice showed profound improvements in this score in both sexes (Fig. 7f, h), resulting in a significant decrease in phenotypical age in male mice (Fig. 7g) which was also seen by trend in female mice (Fig. 7i). Similar to young mice on high fat diet, aged mice performed significantly better during treadmill exercise already after 9 weeks of treatment (Fig. 7j), suggesting isofalcarintriol as potent exercise enhancer. As the endurance capacity was sex independent (Supplementary Fig. 15c), we started to analyze further frailty parameters that were identical in both sexes and depicted them as male and female mice combined (Fig. 7j-m). We found that not only the total frailty index prior the humane endpoint of the individual animal (Fig. 7k) but also two major parameters of well-being, the grimace scale as an indicator of pain (Fig. 7l) and the intensity and/or depth of breathing (Fig. 7m), was improved upon isofalcarintriol treatment in aging animals. Besides that, we have indications that isofalcarintriol might also influence other ocular (Supplementary Fig. 16a–f), physical (Supplementary Fig. 16g–i), cardiovascular, and immunological health parameters at least in one sex. For example, we found that male mice showed increased muscular function as measured *via* the quantification of forelimb grip strength (Fig. 7n). In contrast, female mice displayed an increase in heart rate variability (HRV) (Fig. 7o) as well as an increase in the coefficient of variation (CV) (Fig. 7p), while heartbeat rates were unchanged during electrocardiography (Supplementary Fig. 15e), suggesting an improved cardiovascular function. Moreover, the white blood cell count (WBC) (Fig. 7q) and in particular the number of lymphocytes was significantly decreased in female whole blood samples. Additionally, levels of anti-inflammatory cytokines IL-4 (Fig. 7r) and IL-10 (Fig. 7s) were increased in female plasma samples while pro-inflammatory cytokines were unchanged (Supplementary Table 9), indicating decreased age-related inflammation promoted by isofalcarintriol (**1a**). However, these parameters were not altered in the respective other sex (Supplementary Fig. 15d, f–h), implicating that further studies are essential to validate potential sex-specific health benefits.

Taken together, we have evidence that long-term isofalcarintriol (**1a**) treatment affects glucose metabolism and boosts endurance in both, metabolically challenged young mice as well as aged mice.

Besides that, isofalcarintriol improves several parameters of frailty with increasing age, including measures of physical fitness, discomfort, and respiratory function.

## Discussion

This study identified isofalcarintriol (**1**), a carrot-derived phytochemical as a novel health promoting NRF2 activator and selective inhibitor of the mitochondrial ATP synthase. Evidence for NRF2 activation and mitochondrial ATP synthase-mediated effects manifest in ROS signaling and AMPK activation in cells and *C. elegans*, resulting in stress adaptations such as increased oxidative stress resistance, enhanced mitochondrial biogenesis and permanently increased ATP levels in *C. elegans*. Most strikingly, isofalcarintriol-mediated stress signaling leads to improvement of the metabolic phenotype and endurance exercise performance of mice on high-fat diet, as well as aged mice and has the potential to counteract an age-related increase in frailty. Thus, application of our compound to *C. elegans* and higher organisms has a strong translational potential to ameliorate dysregulations of aging including but not limited to physical fitness, and diabetes.

Aging is characterized by a decay of energy production, impairment of mitochondrial function and an increase in oxidant production[22]. Hence, the mitochondrial ATP synthase as the predominant ATP producer of a cell plays a key role in aging. We observed that concomitantly with a drop of cellular ATP levels and oxygen consumption rates, isofalcarintriol (**1a**) treatment resulted in the transient activation of AMPK signaling by phosphorylation in cells and *C. elegans*. To restore energy levels long-term, we have evidence for an increased glucose sensitivity and blood glucose clearance as seen in glucose tolerance tests in mice on high-fat diet. These metabolic adaptations might possibly be achieved by AMPK-mediated translocation of GLUT4 to the plasma membrane, allowing the cellular uptake of glucose, as seen in in vitro studies using the AMPK agonist AICAR as inducer of AMPK signaling[23] or upon piceatannol supplementation in diabetic mice[24]. Indeed, other polyacetylenes such as falcarindiol and falcarinol were shown to induce glucose uptake in adipocytes and myotubes, indicating a possible involvement of these structures in glucose metabolism[25]. However, whether GLUT4 translocation is part of isofalcarintriol-mediated signaling remains to be examined. In addition to improved glucose metabolism, we observed exercise-like adaptations including an increase in mitochondrial biogenesis and enhanced exercise performance, most likely as direct result of AMPK signaling. Hence, treatment with this natural compound mimics effects of exercise training as previously seen with the AMPK activator AICAR[26]. These adaptations might alleviate frailty in aged organisms, as demonstrated in improved frailty indices in this study. Together with our proteomics data, the phenotypical similarities between piceatannol and isofalcarintriol (**1a**) underline our hypothesis of ATP synthase α subunit as specific molecular target of isofalcarintriol (**1a**), as piceatannol was previously shown to inhibit the F1 ATP synthase by binding in the hydrophobic pocket of an annulus between the α, β, and γ subunits[5]. Indeed, ATP5A has been described as common target against aging and was shown to be downregulated in type-2 diabetes mellitus[27]. Previous publications demonstrated a conserved role of the mitochondrial ATP synthase in the aging process. For example, RNAi-mediated downregulation of ATP5A/*atp-1* and ATPO/*atp-3* has been shown to increase *C. elegans* lifespan by 27% and 21%, respectively[28], thus identifying the ATP synthase as potent target for anti-aging therapies.

We assume that mitohormesis-mediated redox adaptations as seen in this study are directly responsible for the observed isofalcarintriol-dependent exercise mimetic effects and improvements of glucose metabolism, both known to positively affect aging and age-related diseases including neurodegeneration[29,30]. Supporting this, health improvements by physical exercise were shown to be crucially dependent on ROS signaling, as antioxidant treatment of exercising humans prevented induction of molecular mediators of

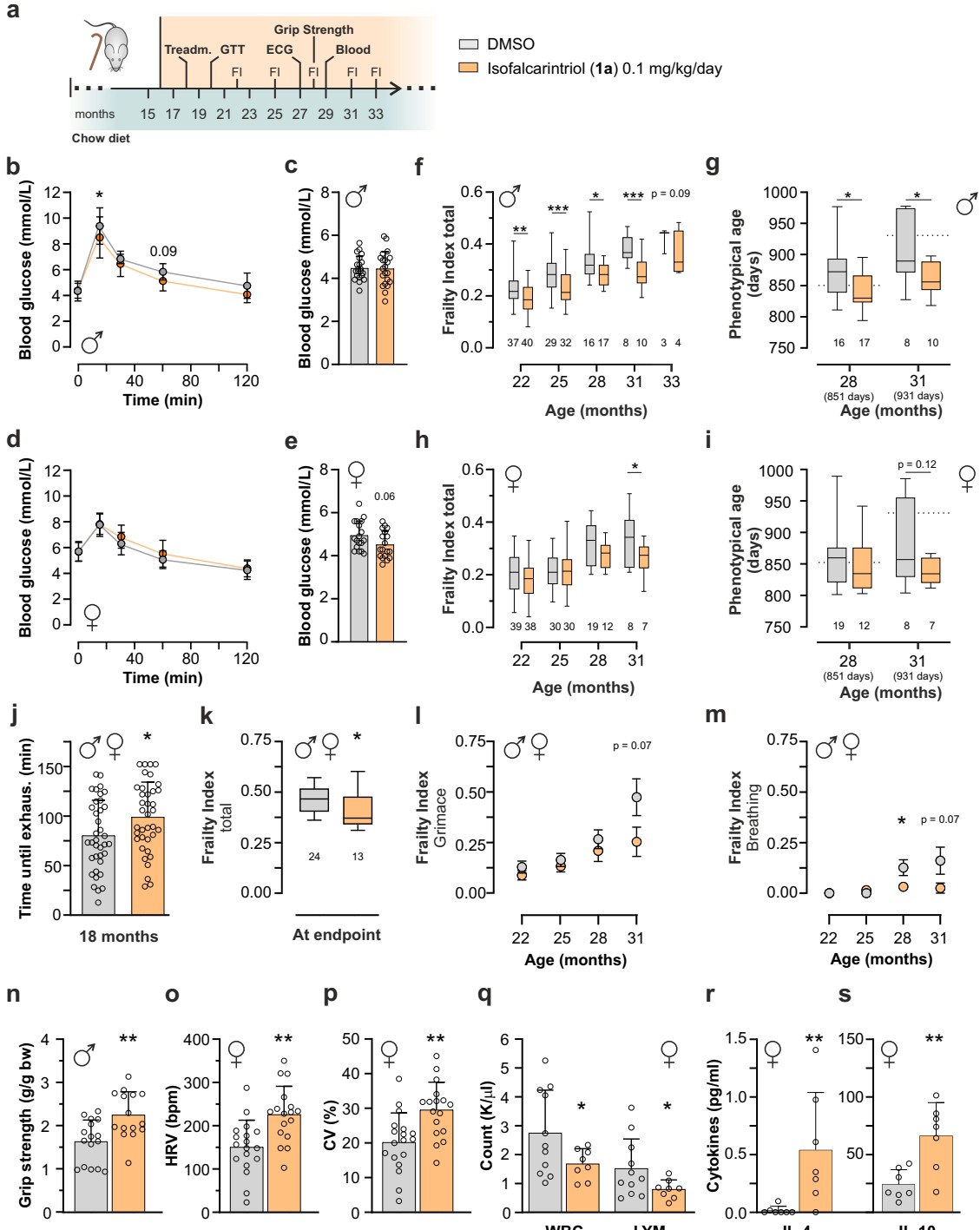

**Fig. 7 | Effects of isofalcarintriol in aged mice on glucose sensitivity, exercise capacity, and parameters of frailty. a** Schematic overview of isofalcarintriol (**1a**) supplementation study with aged C57BL/6NRj mice on chow diet. **b** Glucose tolerance test 20-month-old male mice (DMSO: $n = 10$; IFT: $n = 11$ mice; 15 min: $p = 0.03$) and (**c**) fasted blood glucose levels at 29 months of age ($n = 21$ mice per treatment). **d** Glucose tolerance test in 20-month-old female mice ($n = 12$ per treatment) and (**e**) fasted blood glucose levels at 29 months of age ($n = 18$ mice per treatment). Total frailty index (FI) score of (**f**) male (22 months: $p = 0.0078$; 25 months: $p = 0.0005$; 28 months: $p = 0.0254$, 31 months: $p = 0.0007$) and (**h**) female mice ($p = 0.03$) ($n =$ as indicated below bars). Calculated phenotypical age based on FI scores, (**g**) of male mice (28 months: $p = 0.0169$; 31 months: $p = 0.0246$) and (**i**) of female mice ($n =$ as indicated in panels). Sex-independent frailty parameters, shown as males and females combined, including (**j**) endurance capacity ($p = 0.03$) ($n = 37$ mice per treatment), (**k**) total frailty index score at individual

endpoints ($p = 0.01$), as well as (**l**) grimace and (**m**) breathing scores (28 months: $p = 0.0378$). Sex-dependent health parameters in either male or female mice, including (**n**) grip strength of male mice (month 28) (DMSO: $n = 16$; IFT: $n = 15$ mice) ($p = 0.0036$, **o** heart rate variability (HRV) ($p = 0.0014$) and **p** coefficient of variation (CV) ($p = 0.002$) as measured via electrocardiogram (ECG) in female mice (month 27) (DMSO: $n = 18$; IFT: $n = 17$ mice), **q** White blood cell count (WBC) ($p = 0.04$), and number of lymphocytes (LYM) ($p = 0.048$) (DMSO: $n = 11$; IFT: $n = 8$) as well as concentration of anti-inflammatory cytokines (**r**) IL-4 ($p = 0.0064$) and (**s**) IL-10 ($p = 0.0070$) in female plasma (month 29) ($n = 7$ mice per treatment). Data in (**b**)–(**e**), (**j**), and (**n**)–(**r**) are represented as average + SD or ± SD. Box plots indicate median (middle line), 25th, 75th percentile (box) and min, max values (whiskers). Data in (**l**) and (**m**) show the average ± SEM. Statistics: two-sided unpaired student's *t*-test; two-way repeated measures ANOVA; mixed effects analysis; Mann–Whitney test. Source data are provided as a Source Data file.

insulin sensitivity as well as oxidative stress resistance as shown by us previously[31]. Aged organisms have decreased expression and activity of NRF2[6]. Therefore, increased NRF2 activation, is health beneficial against elevated oxidative stress in elderly and individuals suffering from age-related diseases[32]. Moreover, lipoxidation of ATP5A leads to a loss of enzyme activity and is increasingly occurring in brains of early-stage AD patients compared to age-matched controls[33]. Consequent impairment of energy production might represent an early event in the development of the disease. Targeting the α-subunit of the mitochondrial ATP synthase with a synthetic compound was shown to reduce oxidative stress and Aβ formation and enhance memory and cognitive function in preclinical models of AD[34]. Interestingly, we also have evidence for isofalcarintriol-mediated improvements in our basic protein aggregation models of AD and HD in *C. elegans*, which might be a first indicator of its potential relevance in complex neurodegenerative diseases.

ATP synthase is a key structure in oncogenic transformation due to its involvement in energy metabolism[35]. For instance, the α-subunit of ATP synthase is increasingly expressed in circulating tumor cells and downregulation or inhibition thereof negatively impacts cell migration[36]. Piceatannol, another inhibitor of the F1 ATP synthase[5], was shown to have a similar therapeutic potential against cancer growth and tumor formation[37]. Interestingly, several polyacetylenes, as single molecules or as part of plant extracts, were described to possess anti-cancer properties by acting cytotoxic in multiple tumor cell lines, while non-tumor cells were less vulnerable[38,39]. Similarly, isofalcarintriol (**1a**) inhibited cancer cell proliferation and in vitro colony formation in this study. Thus, it would be of high interest to further characterize its potential as anti-aging agent in advanced follow-up studies.

Despite the existence of known natural compounds exhibiting significant anti-aging effects in model organisms, isofalcarintriol (**1a**) possesses notable advantages over them, such as its versatile bioactivities even at remarkably low doses. Published investigations on anti-aging structures such as resveratrol, curcumin, quercetin, and epicatechin-3 gallate have demonstrated their impact on the lifespan of *C. elegans* at concentrations ranging from 10 to 200 μM (https://genomics.senescence.info/drugs/). In contrast, isofalcarintriol (**1a**) is bioactive at concentrations in the low nanomolar range. Moreover, the biologically effective concentration of resveratrol in mice is 300 mg/kg food, whereas curcumin and green tea extract are administered at 2000 mg/kg food[40], which exceeds the dosage of isofalcarintriol (**1a**) by a factor of hundreds to thousands. Therefore, we anticipate that isofalcarintriol (**1a**) exhibits enhanced bioavailability and/or increased potency as a natural anti-aging agent in contrast to previously documented structures. Furthermore, the exceedingly low bioactive dosage facilitates its inclusion into a natural diet rich in carrots, which has been widely and safely consumed by humans worldwide for centuries. Given the abundant availability of carrots as source material, strategies can be envisaged to enrich our compound in the form of a nutritional supplement that fosters the promotion of healthy aging. In conclusion, the natural polyacetylene isofalcarintriol might be of high interest for further preclinical and clinical evaluation as food supplement or drug to alleviate the negative effects of aging and age-associated pathologies and improve quality of life for the aging human population.

## Methods
### Cell culture conditions
Unless otherwise stated, cells were cultured with DMEM (Sigma Aldrich; #D6429) and 4.5 g/l glucose, phenol red, 10% FBS, and 1% penicillin/streptomycin with pH 7.4 and were incubated at 37 °C, 5% $CO_2$, and 95% relative humidity. Wild-type HepG2 (#300198), HEK293 (#300192), and MCF-7 (#300273) were derived from CLS Cell Lines Service GmbH, HT-29 from Leibniz institute DSMZ (#ACC 299), HepG2 ARE reporter cells from BPS Bioscience (#60513), and HMEpC from

Sigma Aldrich (#830-05 A). HMEpC cells were grown in ready-to use human mammary epithelial cell growth medium (Sigma Aldrich; #815-500).

### Generation of cell lines
The *NRF2* -/- knock-out HEK293 ARE/NRF2 luciferase reporter cell line and Catalase overexpressing HEK293 ARE/NRF2 luciferase reporter cell line were generated by us.

Vectors for CRISPR/Cas9-mediated deletion of 326 bp of human *NRF2* gDNA (part of exon 5) are based on vector pSpCas9(BB)−2A-Puro (PX459) V2.0 (Addgene, #62988) and were cloned according to the detailed protocol provided by the Zhang lab[41]. Top and bottom strand oligonucleotides with 20 bp *NRF2*-specific guide sequences were annealed and inserted into pSpCas9(BB)−2A-Puro (PX459) V2.0 using scarless cloning with the BbsI restriction enzyme. Oligonucleotides were designed to target sequences followed by NGG at their 3'-end in the genomic context (5'-ACTAAACACAAGTCCCAGTG-3' and 5'-TTGT GAGATGAGCCTCCAAG-3', respectively). The resulting vectors were co-transfected into the HEK293 ARE/NRF2 luciferase reporter cell line to obtain the respective reporter with a concomitant *NRF2* loss-of-function mutation. Vectors to overexpress human catalase cDNA ORFs are based on vector pcDNA3.1 + /C-HA, containing a neomycin resistance gene and additionally modified to encode a linker followed by a 6xHis tag prior to the C-terminal HA tag. A sequence encoding the human catalase cDNA ORF without stop codon and the last four amino acids was inserted in-frame with the linker6xHis-HA tag encoding sequence, resulting in a vector to overexpress C-terminally tagged human catalase under control of the CMV promoter.

Stable transfection was conducted in 6-well plates with Lipofectamine 3000 Transfection Reagent (Thermo Fisher Scientific, #L3000008) according to manufacturer's instructions. Transfected cells were split into 15 cm dish culture plates after 24 h and treatment with selection media started after 48 h. The cells were allowed to grow for 7-14 days and single colonies were picked with the help of cloning discs. Homozygous deletion of *NRF2* was confirmed by PCR and catalase overexpression validated via catalase activity assay.

Vectors to overexpress human catalase cDNA ORFs are based on vector pcDNA3.1 + /C-HA, containing a neomycin resistance gene and additionally modified to encode a linker followed by a 6xHis tag prior to the C-terminal HA tag. A sequence encoding the human catalase cDNA ORF without stop codon and the last four amino acids was inserted in-frame with the linker6xHis-HA tag encoding sequence, resulting in a vector to overexpress C-terminally tagged human catalase under control of the CMV promoter.

Stable transgenic cell lines were treated with different antibiotics: HEK293 ARE/NRF2 luciferase reporter cell line with 100 μg/ml hygromycin B, *NRF2* -/- knock-out HEK293 ARE/NRF2 luciferase reporter cell line with 100 μg/ml hygromycin B and 0.25 μg/ml puromycin, HepG2 ARE/NRF2 luciferase reporter cell line with 600 μg/ml G418, and Catalase overexpressing HEK293 ARE/NRF2 luciferase reporter cell line with 100 μg/ml hygromycin B and 200 μg/ml G418.

### Compound library
The compound library (MEGx library) was purchased from AnalytiCon Discovery GmbH, Potsdam, Germany. Validation experiments were performed with synthesized isofalcarintriol (**1a**), 10-gingerol (**2**) (Sigma Aldrich; #G5798) and alnusone (**3**) (ChemFaces; #CFN89536).

### Bacteria preparation
Unless otherwise stated, *C. elegans* was cultured at 20 °C on nematode growth media (NGM) seeded with *E. coli* OP50. Bacteria to feed *C. elegans* were prepared as previously described[42]. Briefly, bacteria were cultured overnight at 37 °C with constant shaking in flasks with appropriate media (OP50 in DYT medium, and E. coli HT115 (DE3) in LB medium containing 100 μg mL−1 Ampicillin). Overnight cultures were

concentrated by centrifugation for 30 min at 3,200 x g and 4 °C. Concentrated bacteria prepared were spotted on NGM plates and left overnight before use.

Heat inactivated (HIT) OP50 were prepared as previously described[43]. In short, the overnight OP50 culture was pelleted by centrifugation as above, all DYT media removed, and bacteria resuspended in S-buffer supplemented with 1 M MgSO4 and 5 mg/ml Cholesterol to have a 10-fold concentrated culture. Afterwards, the bacterial suspension was placed in a 65 °C water bath for 45 min. HIT OP50 were spotted on NGM agar plates on the day of use and dried for 30 min before adding the worms.

### *C. elegans* culture & lifespan

Strains that were obtained from Caenorhabditis Genetics Center (University of Minnesota, USA) included Bristol N2 (wild type), EU31 *skn-1* (*zu135*), RB754 *aak-2* (*ok524*), GMC101 (*dvIs100*). AM23 (rmIs298[pF25B3.3::Q19::CFP]) and AM716 (rmIs284[pF25B3.3::Q67::YFP]) strains were kindly provided by R. I. Morimoto. MIR257 risIs28[hsp-16.2p::CTL1::GFP + unc119(+)] was generated by G.G., and has not been published yet. F27C1.7 (*atp-3*) RNAi was derived from the ORF library v1.1 (Thermo Fisher Scientific) and H28O16.1 (*atp-1*) RNAi from the AHR library (Source BioScience, Nottingham, United Kingdom).

The compound or DMSO (solvent control) was added into liquid hot NGM (50 °C) prior to plate pouring. All compound life spans were performed on heat-inactivated OP50. RNAi-mediated gene knockdown experiments, using E. coli HT115 bacteria feeding, was conducted as previously described[44]. Life span assays with *C. elegans* were performed as previously described, explicitly by omitting FUdR[42], summarized as follows: Adult nematodes were allowed to lay eggs for four to nine hours and the resulting eggs incubated for 64 h at 20 °C on NGM agar plates inoculated with OP50 to obtain a synchronized population of young adult nematodes. For a typical lifespan assay, 100 young adult nematodes per condition were manually transferred to NGM agar plates (30–35 nematodes per 55 mm petri dish) that were inoculated with the respective bacteria as indicated. For the first 10-12 days, nematodes were transferred daily and afterwards every 2-3 days. Nematodes showing no reaction to gentle stimulation were scored as dead. Nematodes that crawled off the plates, displayed internal hatching, or a protruding vulva were censored.

Statistics for lifespan assay are listed in Supplementary Tables 2 and 5.

### ATP Assay

HepG2 cells were seeded into a white clear bottom 96– well plates (Greiner Bio-One, #655098) at 24 h prior treatment. Treatment was performed in at least triplicates by the addition of the 5 μg/ml (initial screening) or 10 μM (validation experiments) of compound, or 10 μM of positive controls including oligomycin (Apollo; #APOBIO1002), piceatannol (Sigma Aldrich; #P0453), and Bz-423 (Sigma Aldrich; #SML1944). In case of *C. elegans* samples, 4M G-HCl was added to previously prepared worm powder, boiled for 15 min and subsequently added to a white clear bottom 96 well plate (Greiner Bio-One, #655098) as 4-fold determination. CellTiter-Glo® Luminescent Cell Viability Assay (Promega; #G7571) was conducted according to manufacturers' instruction. In addition, an ATP standard row was processed in parallel, and the sample protein quantified via Pierce BCA protein assay (ThermoFisher Scientific). The chemiluminescence of ATP standard plates and cell plates was measured with a CLARIOstar microplate reader (BMG LABTECH).

### NRF2 activation

Approximately 50,000 NRF2 Luciferase Reporter cells were seeded into a white clear bottom 96- well plate (Greiner Bio-One, #655098) and incubated overnight. Parallel to this, another 96 well plate was seeded and treated equally for later protein determination via SRB. After 18-22 h, cells were treated with 5 μg/ml (initial screening) or 10 μM (validation experiments) of compound, 10 μM ATP inhibitors oligomycin, piceatannol and Bz-423, or 5 μM positive control (sulforaphane; Sigma Aldrich; #S4441), respectively. The protein plate was taken for SRB assay analysis, while the NRF2 activity assay plate was prepared for read-out according to manufacturers' instructions (ONE-Glo™ Luciferase Assay System; #E6120) with a CLARIOstar microplate reader (BMG LABTECH).

### Proliferation Assay / SRB Assay

Cells were seeded into a 96 well plate and treated with isofalcarintriol (**1a**) the day after. Ice-cold TCA (10%) was used for protein precipitation at desired experimental endpoint. Proteins were stained with 0.4% Sulforhodamine B (SRB) (Santa Cruz; #sc-253615A) in 1% acetic acid, washed, and dissolved in Tris. Absorbance was measured at 510 nm with a CLARIOstar microplate reader (BMG LABTECH).

### Cell staining by using click chemistry

HepG2 cells were seeded onto sterile coverslips in 6 well plates with a density of 24,000 cells per well and incubated for 1–2 days. The cells were then treated with 10 μM isofalcarintriol-alkyne (structure **S25**) overnight. A final concentration of 250 nM MitoTracker deep red (Thermo Fisher Scientific; #M22426) diluted in media was added and plates incubated 30 min at 37 °C/5% CO₂. The cells were fixed with 4% PFA and permeabilized with X. Azide-488 (Click Chemistry Tools #1275-1) in click chemistry buffer was added and incubated for 30 min at 37 °C. The coverslips were then positioned upside down in a drop of DAPI-containing mounting media (Thermo Fisher Scientific; #36966/2 ml) on top of a glass dish. Microscopy pictures were taken with an Olympus FluoView 3000 confocal microscope and analyzed with the software Fiji.

### Mitochondrial membrane potential

HepG2 cells were seeded in a black clear bottom 96- well plate (Greiner Bio-One, #655090). After overnight incubation, the cells were treated with 10 μM compound, solvent control, or FCCP (Cayman; CAY15218-10mg). After that, the cells were stained with 500 nM tetramethylrhodamine (TMRE) (Sigma Aldrich; #87917). Fluorescence was measured at Ex/Em: 540/595 nm with a CLARIOstar microplate reader (BMG LABTECH). Raw data were normalized to TMRE-free cells. Afterwards, protein was quantified by Pierce BCA protein assay (ThermoFisher Scientific).

### Soft agar colony formation assay

This assay was conducted as described previously[45]. In short, 5% (w/v) agar stock solution was prepared in PBS, autoclaved and equilibrated at 50 °C. A 0.5% agar solution was prepared and added as bottom layer to 12 well plates. The cells were detached off the cell culture plate by trypsin and 500-2,000 cells per ml were mixed with compound and liquid agar (final concentration of 0.3%). The mixture was added on top of the bottom layer of agar and allowed to solidify for 30 min at RT. Cell culture media was added on top of each well. The plates were incubated 2-3 weeks and the media was changed every 3-4 days. After that, the colonies were stained with 0.01% crystal violet (Sigma Aldrich; #V5265-500ML) and analyzed with a Leica M165FC microscope with Leica camera DFC 3000 G. The number and size of colonies was calculated with ImageJ and the Plugin ColonyCounter.

### Seahorse XF Cell Mito Stress Assay and Real-Time ATP Rate Assay

The Seahorse XF Cell Mito Stress Assay protocol with *C. elegans* was adapted from previous publication[46]. Specifically, synchronized and treated nematodes were diluted in M9 buffer + 25 μM tetramisole hydrochloride (Sigma Aldrich; #L9756) and distributed in a Seahorse

XF24 culture plate (Agilent; #102340-100) with approx. 40 worms per well. The cartridge plate was loaded with FCCP (Cayman; CAY15218) and 4 mM $NaN_3$ and calibrated at 20 °C. The measurement was performed with an Agilent Seahorse XF24 Analyzer.

For cells, 100 µl HepG2 were seeded with a density of 30,000 cells per 96 well into a Seahorse XF 96 cell culture plate (Agilent; #102416-100). After an overnight incubation, cells were pre-treated with 10 µM of the compound of interest, solvent control, positive control (10 µM piceatannol, 10 µM Bz-423, 5 µM oligomycin) or untreated culture media. The assay protocols were conducted according to manufactures protocol (Agilent) by using Seahorse XFe96 Flux Pak (Agilent; #102416-100). In short, a Seahorse XF96 cartridge plate (Agilent; #102416-100) was calibrated with 200 µl calibration buffer per well and was incubated in a CO2 free incubator at 37 °C overnight. On the day of experiment, the cell culture growth media was removed and cells were washed twice with 150 µl XF DMEM assay medium, pH 7.4 (Agilent; #103680-100) supplemented with 5 mM glucose, 1 mM pyruvate, and 2 mM glutamine., In a last washing step, 150 µl XF DMEM assay medium with (pre-treatment group) or without (injection group) the compound of interest or solvent control was added. In case of the Cell Mito Stress assay, port A was loaded with compound (for injection samples) or with Seahorse medium (for pre-treated cells), port B with oligomycin (Apollo; APOBIO1002), port C with FCCP (Cayman; CAY15218), and port D with a mixture of antimycin A (Sigma Aldrich; A8674-25MG) and rotenone (Sigma Aldrich; #R8875). In case of the Real-Time ATP Rate assay, the injection of FCCP was skipped. The cartridge plate was inserted into the Agilent Seahorse XF96 Analyzer for calibration at 37 °C. After calibration, the cell plate was inserted and the measurement started. After the run, the media was removed and the cell plate incubated with 10 µl NaOH for 1 h at RT for cell lysis and later the protein was quantification quantified via BCA (Pierce BCA protein assay, ThermoFisher Scientific). Raw data were then normalized to µg of protein and analyzed by Agilent Seahorse Analytics.

### ROS quantification by DCF-DA and Amplex Red
25,000 HepG2 cells were seeded in a black clear bottom 96 well plate (Greiner Bio-One, #655090) and incubated overnight. 25 µM DCF-DA (Sigma Aldrich; #D6883) in phenol-free and FBS-free DMEM was added for 45 min at 37 °C. The cells were treated with 10 µM isofalcarintriol (1a), or 1 µM rotenone (Sigma Aldrich; #R8875) for 15 min at 37 °C. Fluorescence was measured at Ex/Em: 485/535 nm with a CLARIOstar microplate reader (BMG LABTECH). Raw data were normalized to cell-free wells and DCF-DA-free cells. ROS in *C. elegans* was measured via Amplex Red Assay as previously published[47]. In short, synchronized worms were treated with 100 µM Amplex Red (Invitrogen, Carlsbad, USA) and 0.2U/ml of horseradish peroxidase in sodium-phosphate buffer for 3 h. Fluorescence was measured at Ex/Em: 544/590 nm with a CLARIOstar microplate reader (BMG LABTECH). For protein normalization, a Pierce BCA protein assay (ThermoFisher Scientific) was conducted.

### Paraquat stress assay
Synchronized L4 stage worms were transferred to isofalcarintriol-containing NGM plates. On day 3 of adulthood, the nematodes were additionally treated with 10 mM paraquat (Sigma Aldrich; #56177). Dead worms were counted in 10-14 h steps and analyzed with JMP.

### Thrashing assay /Huntington assay
Synchronized L4 nematodes were were treated with isofalcarintriol (1a) for 5 days at 20 °C. On day 5 of adulthood 10–20 worms were transferred into that liquid drop and their bending movement was recorded for 30 s with a Leica system (Leica M165FC microscope with Leica camera DFC 3000 G). At least 220 nematodes per condition were analyzed during two independent experiments via automated analysis using the ImageJ plugin "wrMTrck"[48]. The nematode strains AM23 and AM716 display healthy (AM23) or pathological expression (AM716) of poly Q repeats in the huntingtin gene. Synchronized L4 stage nematodes were put on 25 °C to induce protein expression and were treated with isofalcarintriol (1a) (1 nM) for 48 h. At least 250 nematodes per condition were analyzed during two independent thrashing experiments.

### Alzheimer´s assay (GMC 101)
The genetically modified strain GMC 101 (dvIs100 [unc-54p::A-beta-1-42::unc-54 3'-UTR + mtl-2p::GFP]) expresses the human amyloid β peptide. Synchronized L4 nematodes were put on isofalcarintriol (1a) (1 nM) for 24 h. On day 2 of adulthood, they were upshifted to 25 °C to induce protein accumulation. The number of worms with body paralysis was counted every 12 h. For statistics see Supplementary Table 8.

### Real-time PCR
Total DNA for mtDNA quantification was extracted from cells or grinded nematodes by standard proteinase K and phenol–chloroform extraction. mtDNA/nDNA levels were quantified in at least three biological replicates using SYBR Green select master mix (Applied Biosystems) fluorescence on a 96-well format in CFX96 real time system (Biorad). mtDNA/nDNA ratios were calculated by $2 \times 2^{dCT}$ method[49]. Used primers are based on previous publications[49–51] and sequences are provided in Supplementary Table 10.

### Immunoblotting
Cells were lysed in standard protein isolation buffer and disrupted by sonication. Protein was collected by centrifugation and the concentration determined by Pierce BCA protein assay (ThermoFisher Scientific). *C. elegans* samples were processes by grinding before SDS-PAGE separation and protein immunoblotting. Antibodies against phospho-AMPKα (Thr172) (40H9) (Cell signaling; #2535, lot 21) were used in a dilution of 1:1000, AMPKα (Cell signaling; #2532, lot 19) in a dilution of 1:1000 and actin (Sigma Aldrich; #A5060) was used in a dilution of 1:3000. The HRP linked secondary antibody against rabbit (Cell Signaling, #7074 S, lot 30) was used at recommended dilutions. The signal was visualized using Clarity Western ECL Substrate (Biorad).

### Catalase activity assay
To validate the overexpression of human catalase in HEK293 ARE/NRF2 luciferase reporter cells, cells were lysed, sonicated and centrifuged at 12,000 x g for 15 min at 4 °C. The protein supernatant was used for both, protein determination and catalase activity, as described previously[52,53] with minor adaptations[54].

### Biotin Pulldowns and Mass Spectrometry (MS)
For HEK298, 80% confluent cells were harvested by scraping and pelleted by centrifugation (500 x g for 5 min at 4 °C). The cell pellet was dissolved in 250 µl lysis buffer (20 mM HEPES pH 7.3, 50 mM KCl, 5 mM MgCl2, 0.01% NP40, 2 mM NaF, 2 mM Na3VO4) and incubated for 15 min on ice. Samples were disrupted by sonication (3x for 2 s; 50% amplitude). Samples were left on ice for 15 min and afterwards centrifuged at 13,000 x g for 5 min at 4 °C. The protein supernatant was collected and quantified by BCA (Pierce BCA protein assay, ThermoFisher Scientific). 2 µl of a 2 mM stock of biotin-alkene (S12) (negative control) was added to 80 µl Dynabeads™ M-280 Streptavidin (ThermoFisher, #11205D) in lysis buffer and incubated in an end-over-end rotator for 30 min at RT. The lysis buffer was removed from the beads and 500 µg of protein per condition was added to the beads and incubated for 4 h at 4 °C on an end-over-end rotator. Meanwhile, new beads were prepared by including both, S12 (negative control) and biotin-isofalcarintriol (3 S,8 R,9 R)-15 (2 µl of a 2 mM stock concentration per condition). When 4 h have passed, the samples were put on the magnetic holder and the supernatant collected in a new tube. The

freshly prepared bead-compound mix was mixed with the protein supernatant and incubated another 4 h at 4 °C on an end-over-end rotator. After this incubation time, the beads were washed 5 times with lysis buffer without protease inhibitor and without detergent. Dry beads were frozen in liquid nitrogen until further processed.

For HepG2, 3.5 million cells per 10 cm culture dish were seeded and incubated until reaching 80% confluency. Cells HepG2 cells were then treated with 10 μM biotin-isofalcarintriol (3 S,8 R,9 R)-**15** or negative control (biotin-alkene (S12)) for 20 min. Cells were harvested by scraping and pelleted by centrifugation (500 x g for 5 min at 4 °C). The supernatant was removed and the cell pellet was flash-frozen in liquid nitrogen and stored at −80 °C until further needed. Prior to the pull-down assay, the cell pellet was thawed on ice, and dissolved in lysis buffer (20 mM HEPES pH 7.3, 50 mM KCl, 5 mM MgCl2, 0.01% NP40, 2 mM NaF, 2 mM Na3VO4) in a ratio of 4:1 (vol:wt). Afterwards, the samples were disrupted by one freeze and thaw cycle. To degrade nucleic acids, the samples incubated with 0.1% benzonase (250 U/μl) (Sigma Aldrich #E1014-5KU) for 30 min on an end-over-end rotator at 4 °C. After this incubation time, the tubes were centrifuged for 20 min at 16,100 x g at 4 °C and the protein was quantified by BCA and (Pierce BCA protein assay, ThermoFisher Scientific). After determination of the protein concentration, 500 μg of protein per condition was taken and mixed with 40 μl of the Streptavidin beads in lysis buffer by gentle snipping. The samples were incubated on an end-over-end rotator for 4 h at 4 °C to allow binding to the beads. Beads were washed 5 times with lysis buffer without protease inhibitor and detergent. Dry beads were frozen in liquid nitrogen until further processed.

As previously described[43], proteins on beads were eluted and digested by adding 20 μl 50 mM Tris-HCl buffer (pH 8.0) supplemented with 2 M urea, 5 mM DTT and 100 ng Sequencing Grade Trypsin (Promega, #V5111) and incubated for 15 min. Digested protein were reduced by adding more DTT to a final concentration of 10 mM, and alkylated with 3 mM iodoacetamide. The digestion was continued at 32 °C for 6 hours or overnight. pH was checked to ensure that they were within the pH range of 7-9 as it is critical for the optimal trypsinization. All incubation steps were carried out on thermoshaker with gentle shaking at 400 rpm.

The trypsinization was stopped by adding 5% trifluoroacetic acid in several steps until it reached 0.5% TFA, the pH was monitored to ensure it was around. Tryptic peptides were centrifuged at max speed before desalting with self-packed C18 Stage-Tips[55]. Desalted tryptic peptides were vacuum dried and resolubilized in 20 μl 3% acetonitrile, 0.1% formic acid, with the help of 10 min on thermoshaker and sonication each. Samples from three biological replicates were pooled and Four microliters of digested peptides were injected for shotgun liquid chromatography with tandem mass spectrometry (LC-MS/MS) as described below.

All MS experiments were done at the Functional Genomics Center Zurich (FGCZ). As previously described[43], peptide mixtures were separated by reversed phase chromatography using Acquity UPLC M-Class system (Waters Inc.) on HSS C18 T3 Col 100 A column (1.8 μm, 75 μm x 250 mm, Waters Inc.). Peptides were separated on a multistep acetonitrile gradient (5–35% in 135 min, 40% in 5 min, and 80% in 1 min) with 0.1% formic acid at a nanoflow rate of 300 nl/min. Eluting peptides were directly ionized by electrospray ionization Orbitrap Fusion Lumos mass spectrometer (Thermo Fisher Scientific) equipped with a Digital PicoView nanospray source (New Objective) in a data-dependent mode.

All raw data were further analyzed with MaxQuant software suite version 1.6.3.3 (Max Planck Institute of Biochemistry, Munich) supported by the Andromeda search engine[56]. Data were searched against a UniProt Human proteime database encompassing 75,004 protein entries, downloaded from UniProt (Proteome ID: UP000005640.

Peptides were searched with carbamidomethylation as a fixed modification and protein N-terminal acetylation and methionine oxidation as variable modifications. A maximum of two or four missed cleavages were allowed while requiring strict trypsin specificity, and only peptides with a minimum sequence length of six were considered for further data analysis. Data were search with concatenated target/decoy (forward and reversed) version of the libraries. Only proteins identified with at least 2 peptides were included for quantification. Proteins with ≥1.5-fold intensity over negative control were considered as the interaction partner.

The mass spectrometry proteomics data have been deposited to the ProteomeXchange Consortium via the PRIDE[57] partner repository (https://www.ebi.ac.uk/pride/) with the dataset identifier PXD037671.

## Mouse housing conditions

Mouse experiments were performed according to Art.18 Tierschutzgesetz (TSchG), Art. 141 Tierschutzverordnung (TschV), Art. 30 Tierversuchsverordnung (TVV) (all Switzerland) and were approved by the cantonal veterinary office Zürich, Switzerland. The studies were carried out on male and female C57BL/6NRj mice (Janvier Labs). The animals were housed in a specific pathogen-free environment at 22 °C with a reverse 12 h/12 h light/dark cycle with ad libitum excess to water and food. The animals were either fed with chow diet (young animals: Granovit; #10343700PXL15), aged animals: Ssniff Spezialdiäten GmbH; #S8022-S005) or young animals on high fat diet (Ssniff Spezialdiäten GmbH; #E15744-34; 45 kJ% fat). In case of the latter, 6-week-old mice were put on high-fat diet 6 weeks before the start of compound treatment until their sacrifice. Compound supplementation of young mice started at 13 weeks of age by applying 0.1 mg/kg body weight per day of isofalcarintriol (**1a**) or solvent control via autoclaved drinking water. Aged mice were treated from 16 months of age onwards using the same dose as the young mice (0.1 mg/kg isofalcarintriol or solvent control) in autoclaved drinking water. The drinking bottles were replaced weekly. All animals were sacrificed by cervical dislocation at the end of the respective study, except for the aging study, where mice were inspected 4 times per week for health status, and natural deaths were recorded, while moribund animals were sacrificed by cervical dislocation, and recorded as dead. Criteria for sacrifice included progressive body weight loss, unresponsiveness, persistent abnormalities in breathing, large tumor formation, and strong evidence of discomfort and pain.

## Treadmill acclimatization and endurance capacity

The mice were familiarized with the treadmill (Panlab/Harvard Apparatus, Holliston, MA) in 4 single training sessions. In case of lacking motivation to run, the animals were motivated with air puffs. Prior to the experiment, a 5 min warm-up at 8 m/min and 10% incline was performed. The endurance capacity experiment for the high-fat diet cohort (29 weeks of age) was conducted with 5% incline at 12-15 min/min until exhaustion or a maximum of 2 h and 2.5 h for males and females, respectively. The endurance capacity experiment with aged mice (18 months of age) was performed with 10% incline at 12-15 min/min until exhaustion or a maximum of 2.5 h.

## Metabolic, body composition and blood parameters

Experiments were performed as described previously[58] with some adaptations. In short, fasted blood was sampled via submandibular bleeding and collected in heparin-containing tubes. Blood glucose was determined using a hand-held glucose meter (Bayer Contour XT Meter). Plasma free fatty acids (FFA), triglycerides (TGs), Cholesterol, and HDL, were analyzed by enzymatic reaction (Cobas Mira; La Roche, Basel, Switzerland). Hematology parameters were measured in a sample of whole blood collected at 29 months of age. The sample was run in a Hemavet machine (Drew Scientific, Miami Lakes FL, USA) according to the manufacturer instructions. Parameters measured included white blood cells, neutrophils, lymphocytes,

eosinophils, basophils, platelets, hemoglobin, hematocrit. Cytokines in blood plasma samples were processed with the V-PLEX Proinflammatory Panel 1 Mouse Kit (MSD, Rockville, MD, USA) and analyzed with a Sector Imager Microplate Reader (MSD, Rockville, MD, USA) according to manufacturer's instructions.

Glucose tolerance tests (GTT) were performed in 4-h-fasted mice (15 weeks of age in high-fat diet cohort and 20 months of age in aging cohort), by intraperitoneally injecting a bolus of D-glucose (1-2 mg g$^{-1}$). PhenoMaster (TSE Systems, Bad Homburg, Germany) open-circuit calorimetry system was used to measure oxygen consumption and carbon dioxide production over several days following a 24-h acclimation period. Analysis was done via CalR, a web-based analysis tool for calorimetry experiments[59]. Body composition was measured by nuclear magnetic resonance (Echo MRI-100 Body Composition Analyzer; Echo Medical Systems, Huston, TX, USA) as described previously[60].

### Frailty Index, Grip strength, and Electrocardiogram (ECG)
The frailty index was determined between 22-33 months of age as previously described[20] by analyzing 31 non-invasive parameters of aging and the phenotypical age was calculated via machine learning analysis at http://frailtyclocks.sinclairlab.org/[21].

Forelimb grip strength was measured via grip strength test meter (Bioseb in vivo Research Instrument) according to manufacturer's protocol with some adaptations as reported previously. Three measurements per animal (28 months of age) were averaged and normalized to total body mass[61].

ECG was measured with ECGenie Machine (Mouse Specifics, Boston MA, USA) according to manufacturer's protocol by using EMouse Software (Mouse Specifics, Boston MA, USA) and LabChart software (ADI Instruments, Bella Vista NSW, Australia). In short, the mice (27 months of age) were allowed to acclimatize for up to 10 min on the arms of the apparatus. After that, the recording was started for up to 20 minutes depending on the calmness of the animal and the quality of recording. 3-5 recordings per animal were averaged to get a representative reading.

### Reporting summary
Further information on research design is available in the Nature Portfolio Reporting Summary linked to this article.

## Data availability
The mass spectrometry proteomics data in this study have been deposited to the ProteomeXchange Consortium via the PRIDE[57] partner repository (https://www.ebi.ac.uk/pride/) with the dataset identifier PXD037671. All data generated in this study are provided in the Supplementary Information and the Source Data file. Source data are provided with this paper.

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

## Acknowledgements

*C. elegans* strains used in this work were provided by the Caenorhabditis Genetics Centre (Univ. of Minnesota, USA), which is funded by NIH Office of Research Infrastructure Programs (P40 OD010440). The AM23 and AM716 strains were a kind gift of R. I. Morimoto. Furthermore, we acknowledge the contribution of T. Karakaya in generating two cell lines also used in this work, as well as D. Pöhlmann and B. Laube for their excellent technical support. We thank Functional Genomics Center Zurich (FGCZ) for support in acquisition and analysis of mass spectrometry data. The authors thank Dr. M.-O. Ebert, R. Arnold, S. Burkhardt, and R. Frankenstein for NMR measurements and Dr. B. Rubi, L. Bertschi, M. Meier, and D. Wirz for HRMS measurements and analysis. This work was funded by the Swiss National Science Foundation (Schweizerischer Nationalfonds, SNF 31003A_156031, 31003A_176127, and 310030_204511, all to M.R.), and ETH Zürich.

## Author contributions

C.T., F.F and M.R. conceived the project; C.T, R.E., E.M.C. and M.R. designed the research; C.T performed and analyzed the majority of biological experiments with additional experiments performed by F.F., K.Z., G.G., G.L., and J.Y.W., as well as analyses by J.Y.W and S.J.M; R.E. and E.M.C designed, and R.E. performed and analyzed all chemical experiments and synthesis; C.T. prepared all biological figures, and manuscript text; R.E. prepared all chemical figures, and manuscript text; E.M.C and M.R contributed in writing the manuscript.

## Funding

## Competing interests
Data presented in this manuscript are contained in the patent application WO2023143988A1 with ETH Zürich as applicant, and E.M.C., M.R., C.T., and R.E. as inventors. E.M.C., M.R., C.T., and R.E. declare no additional competing interest. The remaining authors declare no competing interests.
