## [Peer Review File · Nature Communications]

A naturally occurring polyacetylene isolated from carrots promotes health and delays signatures of agingREVIEWER COMMENTS

Reviewer #1 (Remarks to the Author):

Carolin Thomas et al: A novel naturally occurring polyacetylene isolated from carrots promotes health and delays signatures of aging

They identify isofalcarintriol (ISF), a carrot-derived phytochemical as a novel health promoting

Nrf2 activator and selective inhibitor of the mitochondrial ATP synthase. Evidence for Nrf2 activation and mitochondrial ATP synthase-mediated effects manifest in ROS signaling and AMPK activation in cells and *C. elegans*, resulting in stress adaptations such as increased oxidative stress resistance and somewhat enhanced mitochondrial biogenesis in

C. elegans. ISF-mediated stress signaling leads to improvement of the metabolic phenotype and endurance exercise performance of mice on high-fat diet, as well as mice and has the potential to counteract an age-related increase in frailty.

This is well written paper with extensive identification and experimentation on this compound using worms, cells and mice.

It is not a premise for healthy aging that energy should be targeted so their selection of this compound may not be the best approach in countering age associated defects.

Nevertheless, they do identify something of use and interest.

The effects they see on age associated parameters are significant but not dramatic, and it is not clear why this intervention is more useful than so many other natural substances that have been reported to slow aging. What is missing in the work is a discussion relating this to such other interventions and why this should be preferred. Many other natural substances increase the life span in mice, but here they see no effect.

Also, importantly, there is no safety results. Safety parameters, what was done? DNA damage, Senescence and other standard measures should be done in cells or mice.

There are a number of overinterpretations that should be either further experimentally explored, or the discussion toned down. This includes the mitochondrial parameters, Fig 4.

Some of the mitochondrial parameters are only minimally affected by the compound

The effect on ATP production. They identify this compound as an inhibitor of ATP but after

10 min ATP is minimally decreased and then increases at 48 hrs, Fig 3H

Fig S1. In E, Fig says 0.1 nM but legend says 1 nm ? And why no effect on lifespan at 10 nm ?

Fig 3H. ATP levels were slightly decreased after 10 nM but not after 1 nM in c.elegans

Fig 4. In Fig 4E the maximal respiration should be corrected for differences in baseline. The effects of ISF in fig 4B and C are rather minimal

Reply to Reviewers, NCOMMS-23-20515, Thomas, Erni *et al.*

(Authors' replies in red)

5 Reviewer 1:

They identify isofalcarintriol (ISF), a carrot-derived phytochemical as a novel health promoting Nrf2
activator and selective inhibitor of the mitochondrial ATP synthase. Evidence for Nrf2 activation and
mitochondrial ATP synthase-mediated effects manifest in ROS signaling and AMPK activation in cells
and *C. elegans*, resulting in stress adaptations such as increased oxidative stress resistance and
somewhat enhanced mitochondrial biogenesis in *C. elegans*. ISF-mediated stress signaling leads to
improvement of the metabolic phenotype and endurance exercise performance of mice on high-fat
diet, as well as mice and has the potential to counteract an age-related increase in frailty.

This is well written paper with extensive identification and experimentation on this compound using
worms, cells and mice.

We thank the reviewer for their positive and supportive comments.

It is not a premise for healthy aging that energy should be targeted so their selection of this compound
may not be the best approach in countering age associated defects. Nevertheless, they do identify
something of use and interest.

The most long-standing (and hence probably best studied) intervention to delay aging is calorie
restriction, first described by Clive McCay in 1937, as reviewed in reference 3 of the current
manuscript. Caloric restriction improves health by decreasing cardiac risk factors, improving insulin
sensitivity, and mitochondrial function thereby contributing to healthy aging (ref. 3). Besides that,
physical exercise promotes healthy aging by increasing physical endurance, cardiovascular health, and
metabolic function through an initial energy depletion during the exercise session. Consequently,
targeting energy metabolism by e.g. phytochemicals is in our opinion a very valid premise for healthy
aging, mimicking both calorie restriction as well as physical exercise.

The effects they see on age associated parameters are significant but not dramatic, and it is not clear
why this intervention is more useful than so many other natural substances that have been reported
to slow aging. What is missing in the work is a discussion relating this to such other interventions and
why this should be preferred.

We thank the reviewer for their valuable comment. While there are indeed other natural
compounds known to extend lifespan and promote health in model organisms, and possibly humans, it
should be noted that (i) IFT is similarly effective, (ii) IFT uses a different molecular mode of action, which
(iii) implies that IFT could be combined with other natural compounds to exert additive effects, (iv) is
readily available from carrots and particularly carrot peel as a by-product of carrot processing, and
most importantly (iv) IFT exerts biological activities at unusual low doses.

The Discussion section has been extended accordingly, now stating: *'Despite the existence of*
*known natural compounds exhibiting significant anti-aging effects in model organisms, isofalcarintriol*

*possesses several notable advantages over them, such as its versatile bioactivities even at remarkably*
*low doses. Published investigations on anti-aging structures such as resveratrol, curcumin, quercetin,*
*and epicatechin-3 gallate have demonstrated their impact on the lifespan of C. elegans at*
*concentrations ranging from 10 to 200 μ M (as sourced from <https://genomics.senescence.info/drugs/>).*
*In contrast, isofalcarintriol exhibits lifespan-extending properties at concentrations in the low*
*nanomolar range. Additionally, the bioactivity of resveratrol has been observed at concentrations of*
*300 mg/kg food which is 375-fold higher than the effective concentration of isofalcarintriol in mice (0.8*
*mg/kg water corresponding to 0.1 mg/kg body weight). Similarly, curcumin and green tea extract were*
*administered at concentrations of 2000 mg/kg food (PubMedID: 22451473), surpassing the dosage of*
*isofalcarintriol by a factor of 2,500 (PubMedID: 22451473).*

*Therefore, we anticipate that isofalcarintriol exhibits enhanced bioavailability and/or increased*
*potency as a natural anti-aging agent, different from previously documented structures. Furthermore,*
*the exceedingly low bioactive dosage facilitates its inclusion into a natural diet rich in carrots, which*
*has been widely and safely consumed by humans worldwide for centuries. Given the abundant*
*availability of carrots as source material, strategies can be envisaged to enrich our compound in the*
*form of a nutritional supplement that fosters the promotion of healthy aging.'*

Many other natural substances increase the life span in mice, but here they see no effect.

The National Institute on Aging Interventions Testing Program (ITP) has analyzed natural
compounds shown to extend lifespan in lower organisms, including resveratrol, green tea extract
(mainly including EGCG), curcumin, oxaloacetic acid, and lastly medium-chain triglycerides (PubMedID:
22451473). However, none of the tested compounds was lifespan extending in neither male nor
female mice. Consequently, and congruent with an increasing number of fellow researchers, we
believe that versatile effects on health and age-related frailty parameters are a more precise predictor
for healthy aging than pure lifespan analysis.

Independently, there are examples for compounds which do extend lifespan in specific
laboratories when applied to inbred mouse lines, while showing no effects in the ITP setting, employing
a genetically heterogenous background.

Also, importantly, there is no safety results. Safety parameters, what was done? DNA damage,
Senescence and other standard measures should be done in cells or mice.

We performed a short-term dose finding and toxicity study in young WT mice where IFT was
administered ad libitum in the drinking water over a period of 2 weeks. We included three
concentrations, 0.025, 0.25, and 2.5 mg/kg body weight. These concentrations were far below toxic
doses of other, previously studied polyacetylenes. For example, a single oral gavage of 300 mg/kg body
weight of Falcarinol, a structurally related polyacetylene, did previously not show any signs of toxicity
according to published data (PubMedID: 32905447).

We have added the above as an additional information in line 302-305 and line 310 of the main
manuscript, and have added Fig. S10A,B and S12E-F to the supplementary information: *'Most*
*importantly, a prior two-week toxicity study in animals on chow diet did not show any elevation of liver*
*toxicity parameter such as ALAT and ASAT levels in the blood when applying a maximum dose of 2.5*
*mg/kg isofalcarintriol (Fig. S10A, B), assuming that long-term treatment with the compound in lower*
*doses is save as well. The high-fat diet applied in this study serves to induce overweight and challenge*
*the metabolic phenotype, provoking a decrease in glucose sensitivity and promoting insulin resistance*

as typically seen in obese and diabetic individuals. During the 6 months course of this high-fat diet
study, no major differences in body mass (Fig. S12A), body composition (fat mass and lean mass) (Fig.
S12B), blood lipid levels (Fig. S12C, D), and liver toxicity parameters (Fig. S12E, F) were detected
between treated and control animals.'

In addition, we had analyzed the occurrence of tumors by necropsy of 2+ year old animals that
have been on IFT treatment (0.1 mg/kg) for at least 1 year. As a result, we did not observe any increase
in the number of formed tumors (pls see **Review Figure 1**). As depicted in Fig. S13, IFT-treated animals
had a similar life expectancy as control animals, indicating no major safety concerns of the compound.

**Review Figure 1: Number of tumors at endpoint in aged mice on IFT or DMSO**

There are a number of overinterpretations that should be either further experimentally explored, or
the discussion toned down. This includes the mitochondrial parameters, Fig 4. Some of the
mitochondrial parameters are only minimally affected by the compound.

We thank the reviewer for their comment, giving us the opportunity to further elaborate on
the primary concept of the initial screen (Fig. 1B). As can be seen in this figure, the initial screen was
intentionally performed for compounds that reduce cellular ATP content to a limited extent only. This
was based on the fact that a more prominent reduction in ATP would induce apoptosis and hence
induce toxic side effects in cells, nematodes and mice.

Those mildly ATP reducing compounds (Fig. 1B) were further analysed for NRF2 activation
(Figs. 1D and E). After having established IFT as an inhibitor of ATP synthase (Fig.3) previously
established ATP inhibitors (Bz-423, Piceatannol) were added to the experiments depicted in Figs. 1B,
D and E to test for comparable effects, also on lifespan (Figs. 3E and F).

Very consistently and as to be expected, the experiments in Fig. 4 indeed depict a limited (while
statistically highly significant) effect of the corresponding compounds, including IFT. This, however,
reflects the purpose of the screening design: only a limited impairment of mitochondrial activity will
promoted lifespan, whereas a pronounced inhibition of the electron transfer chain (ETC) will cause
reduced lifespan and/or lethality. This reflects a no-linear ('hormetic') dose-response mechanism,
where limited ETC inhibition promotes longevity, while strong inhibition of the ETC causes increased
mortality. One of the senior authors (M.R.) has shown this repeatedly for many interventions in the
past, and the mechanism is known as mitochondrial hormesis or mitohormesis.

The effect on ATP production. They identify this compound as an inhibitor of ATP but after 10 min ATP
is minimally decreased and then increases at 48 hrs, Fig 3H

This indeed is an essential part of the biochemical mechanism. As also for e.g. health-
promoting exercise, IFT causes a reduction of ATP for a limited period of time only. The initial reduction
is sensed by AMP-dependent protein kinase (AMPK/aak-2 in *C. elegans*; Figs.3L, M) to compensatorily
induce mitochondrial biogenesis (Figs. 6F-H) which leads to a secondary increase in ATP levels at later
time points, paralleled by increased exercise capacity (which is based on mitochondrial mass) (Figs. 6
and 7).

Fig S1. in E, Fig says 0.1 nM but legend says 1 nm ? And why no effect on lifespan at 10 nm?

We thank the reviewer for bringing this inconsistent labeling to our attention. Fig.S1E indeed
depicts a lifespan assay using IFT at the concentration of 0.1 nM (concentration at the panel has been
corrected; concentration in the legend remains).

Regarding the second question: Panel F has depicted (and still depicts) IFT at an concentration
fo 10 nM. What the reviewer possibly refers to, are panels C and D? These depict experiments using
Alnusone, where independent experiments showed inconsistent results. Such inconsistencies rarely
occur in lifespan assays, but have not been observed in any other independent repetition throughout
the manuscript.

Lastly, it should be noted that all lifespan data and statistics are listed in Table S6.

Fig 3H. ATP levels were slightly decreased after 10 nM but not after 1 nM in *c.elegans*

We do not necessarily expect significant changes at lower doses in whole-body lysates of
nematodes. Rather, reductions in ATP are to be expected in specific tissues of the worm, which cannot
be analyzed individually for technical reasons. For this reason and indeed, ATP levels were significantly
decreased at 10 nM only but not after 1 nM in *C. elegans* using three replicates containing pooled
whole worm samples; however, and notably, ATP levels were decreased in *C. elegans* in the
experiments shown in Fig. 4G. Taken together, we believe there is ample evidence for a transient
decrease in ATP levels, also when taking the epistatic dependency on AMPK/aak-2 into account (pls
see above).

Fig 4. In Fig 4E the maximal respiration should be corrected for differences in baseline. The effects of
ISF in Figs. 4B and C are rather minimal

As already stated above, the limited effects of IFT are reassuring since mechanistically
essential, rather than being a reason to be concerned.

To our knowledge, and also according to the Agilent Seahorse Cell Mito Stress user manual
([https://www.agilent.com/cs/library/usermanuals/public/XF_Cell_Mito_Stress_Test_Kit_User_Guide](https://www.agilent.com/cs/library/usermanuals/public/XF_Cell_Mito_Stress_Test_Kit_User_Guide.pdf)
.pdf), the maximal respiration is calculated by subtraction of the non-mitochondrial oxygen
consumption rate after injection of rotenone and antimycin A (see Figure below). Maximal respiration

rates represented in Fig 4 were calculated as suggested there. Corrections for differences in baseline
would result in the calculation of the spare capacity which was not the aim here.

Moreover, the effects of IFT in Fig. 4B and C are in a similar magnitude – or even stronger –
than the established ATP synthase inhibitor piceatannol (PubMedID: 10425214) and Bz-423
(PubMedID: 30022951). Consequently, the effect of IFT is rather profound in our opinion.

Taken together, we believe that we have addressed the reviewer’s questions, and/or have
implemented their valuable suggestions, and think that the revised version of the manuscript may now
be acceptable for publication. Sincerely, we hope that the reviewer will share our view.

Reviewer 2:

The 1,200 single compounds were purchased from a company for this study in the first step of
bioactivity screening on ATP levels in HepG2 cells and Nrf2 activation in HEK293. Why used a cancer
cell line HepG2, not a normal cell line such as HEK293 to measure the compound effect on ATP levels,
as the ROS levels or/and mitochondria bioenergetic status are believed to be different to normal cell
lines?

We thank the reviewer for this very relevant question.

Firstly, we wish to point out that the cell line used primarily served to perform the majority of
experiments, HepG2, is very widely used for metabolism-related cell culture studies. This is reflected
by the fact that the search term "HepG2" generates 812,000 entries in GoogleScholar. Importantly, the
well-established metabolic flexibility of HepG2 cells make them the ideal tool to study changes in
mitochondrial capacity, as performed in the current study (Fig. 1B and others).

By contrast, number of GoogleScholar entries for the search term ("HEK293" AND "HEK-293") is
64,700 combined, i.e. less than 8% of the corresponding HepG2 entries. This may be due to the fact
that HEK cells are embryonic, and (more importantly) carry an unstable chromosome set, as described
repeatedly as ‘hypotriploid’, with increasing loss of chromosome during passaging. Hence, the main
advantage of HEK293 cells is their transfectability, especially with reporter constructs as used in the
current manuscript (Figs. 1C and D).

Thirdly and in addition to their impaired metabolic flexibility, HEK293 cells cannot be used in
Seahorse-based experiments, due to their almost non-adhesive nature of growth in cultured media.

Lastly and most importantly, all findings in HepG2 and HEK293 cells have been fully replicated and
validated in nematodes, and (where possible) in mice. This indicates that the cell-based findings
presented are not cell-line specific, but rather can be translated into living organisms.

If it was a structurally unknown compound, how it was known as a diacetylene as described in line
114? On the other hand, isofalcarintriol is indicated/labeled as 1, 1a, or IFT in Figure 1 and Figure S1
that is somewhat confusing. Whether the data in both figures were obtained using commercial
compound 1 or asymmetry-synthesized 1a that should be clearly described, as the compound purity
was not the same in 1 and 1a (Figure S3 and Table S3).

The compound was contained in a commercially obtained library. The compounds contained
were isolated from plant extracts. Regarding the compound of interest in this manuscript, the supplier
of the library (AnalytiCon Discovery) did not provide a name or C.A.S. number, however did provide a
structure, based on their in-house analyses.

We apologize for the inconsistencies in the usage of the commercially obtained compound '(1)'
and the asymmetry-synthesized compound '(1a)', and have corrected this accordingly.

In brief, only the initial screening experiments depicted in Figs. 1B, D and E were performed
using IFT (1), while all subsequent experiments were performed with the synthesized IFT (1a).

Figure S1: Typo in panel C or D as which data is for 1 nM alnusone treatment? Why 10- gingerol did
not test with 10 nM, and why isofalcarintriol was tested with 0.1 nM, not 1 nM in this experiment?

The labeling was corrected (pls see comment to Reviewer #1), and panels were adjusted so
that they all show 10 nM concentrations. Alnusone and Gingerol were both tested in three
concentrations (1nM, 10nM, 100 nM). It should be noted that all lifespan data and statistics are listed
in Table S6.

Panels C and D **both** depict experiments using Alnusone at 10 nM, where independent
experiments showed inconsistent results. Such inconsistencies rarely occur in lifespan assays, but have
not been observed in any other independent repetition throughout the manuscript. We included both
data sets to make them accessible to the reviewers and future readers.

Lastly, the requested lifespan using 1 nM isofalcarintriol (1a) was (and still is) depicted in the
main Fig. 1.

Figure 1: The name of HEK293 cell line should be indicated in the legend to panel G. It is also suggested
to switch panel G data to panel E following panel D in Figure 1 that would be easier to compare and
understand both data sets.

We fully agree. The sequence of the figure panels as well as the adaptation of the figure legend
has been changed accordingly.

5. There are some flaws in the compound structure characterization and elucidation in this study. The
authors described significant impurities contained in compound 1 (Table S3) and were not able to
assign some spectra data for the compound. Then, in Figure 2A, how the correlation data assigned in
the compound could be given only based on the HMBC data without the support of NOESY and HBQC
analysis/data. In Figure S3, some impurities were observed in the eSFC chromatograms for 1 (A), ent-
1b (B), ent-1a (C), 1b (E); in addition, only selected retention times (10 to 18 min) are shown in Figure
S3. It is not clear whether there are other impurities/peaks present in any chromatogram out of the
time range. The bioactivity data can not be directly compared within impure compounds.

Please accept our apologies for essential parts of the **Supplemental Information (SI)** missing
in the primary submission. It appears that, due to data volume / size issues this (originally separate
file) has not been processed during upload of the original manuscript. We have now resized the file,
and included it into the revised version of the Supplemental Information (SI), specifically **pages 36 to**
**210 (sic)**.

Accordingly, the initially missing spectra for isofalcarintriol (**1, NP017896**) can now be found in
the SI (Spectra to Configurational Assignment of Isofalcarintriol) including ¹H, ¹³C, HSQC, HMBC, COSY
NMR spectra. Full length eSFC chromatograms can be found in the SI (SFC Data to Configurational
Assignment of Isofalcarintriol). The initial screening was conducted with isofalcarintriol (**1, NP017896**)
as provided by AnalytiCon Discovery GmbH and follow-up biological evaluation was conducted with
synthetic isofalcarintriol (**1a**).

6. All the NMR spectral profiles of major synthesized compounds in this study should be provided along
with the spectral data in the supplementary information. All the anti-1,2- diol structure drawing is not
accurate that should be corrected.

Again (pls. see above), please accept our apologies for essential parts of the **Supplemental**
**Information (SI)** missing.

The full synthetic descriptions as well as spectral data have been amended to the SI file in total
190 pages have been added. Regarding the anti-1,2- diol structures: During the synthesis the
compounds are cyclic 4,5 disubstituted 2,2-dimethyl-1,3-dioxolanes (e.g. *anti*-alkyne 4a). After
deprotection the compounds are converted to the corresponding linear compounds which are *syn*-
configured. The linear compounds are not drawn in their fully extended zig-zag configuration to keep
them in line with their precursor structures.

Please uniform all the compound numbers with bold-faced throughout the text.

As stated above and also applicable here, we made the adaptations in the text accordingly.

Figure 2 and Figure S2: Any explanation for why 100 μ M 1a, 1b, 1c and 1d compound treatments
showed much less fold activation of Nrf2 compared to their 10 μ M treatments?

As shown both functionally and by mass spec in this manuscript, IFT is an ATP synthase inhibitor
which, as all such inhibitors, eventually will impair growth and induced apoptosis. Accordingly, the
reduced activation might be explained by the toxicity of the compound at high concentrations leading
to a decrease in cell number (see **Review Figure 2**, below), and most likely also to significant changes
in cellular signaling pathways, thus resulting in attenuated Nrf2 signaling.

**Review Fig. 2: Cell protein content upon treatment with compound variants at 10 vs. 100 μM**

Figures S4 and S5: In these two figures, the importance of the C10 to C17 moiety and the hydroxy
 group at C3 in the compound structure were demonstrated. However, the biotin-2 isofalcarintriol (15)
 was inactive in HEK293 cells, conversely, it retained some activity in HepG2 cancer cells. What made
 this bioactivity discrepancy in both cell lines? In addition, what is the rationale to use an inactive
 compound probe to do protein pulldown experiment for their specific targeting/binding protein(s) in
 HEK293 cells?

As outlined in lines 171-174, as well as in the Supplementary Information, some cell lines
 (including HepG2) express specific biotin transporters in their plasma membrane which enables them
 to take up biotin as well as biotin-labelled compounds. However, and by contrast, HEK293 cells lack
 such transporters, and thus are unable to take up biotin-labelled compounds (pls see for details
 PubMedID 26021457).

This difference explains why biotin-isofalcarintriol appears to be inactive in intact HEK293, i.e.
 solely due to the lack of cellular uptake. Given that information, the biotin pulldown protocols were
 adapted accordingly: While biotin-IFT treatment was performed in intact HepG2 cells, where its uptake
 is possible, the treatment of HEK293 was performed in lysed cells, allowing the compound to access
 proteins directly (pls see Fig. S6).

Figure 3: In panel K, it needs to provide the cell image treated with azide488 probe alone to compare
 with the cells treated with IFT-azide488 conjugate.

Indeed, and we wish to apologize for suboptimal labeling: Both the “DMSO” solvent control
 and “IFT-alkyne” condition in Fig. 3K were all co-treated with azide488. The figure legend was adapted
 accordingly, to avoid further misunderstanding.

The concentrations used for both cell lines and C. elegans assays have 1000-fold differences (μM vs.
 nM), please explain the reason. Please delete one repeated word “including” in line 832.

This is a well-known issue, which the authors’ biological lab (M.R.) as well as many other labs
 have experienced for a significant number of different compounds.

E.g. in a previous publication on arsenite we observed similarly high differences in effective
compound concentrations between cells and *C. elegans* (PubMedID 23534459). Arsenite was used in
concentrations up to 100 μ M in HepG2 cells. By contrast, Arsenite was lifespan-extending in a 1000-
300 fold lower concentration in *C. elegans* (100 nM).

This aligns with the observed variations in concentration as outlined in this manuscript (μ M in
cells vs nM in *C. elegans*). This discrepancy might result from differences in uptake, distribution, and
metabolism of the compound (cellular vs organismal).

11. Table S4: The experiment to analyze the spatial distribution of 1a in *D. carota* should be described
with details, such as the extraction conditions (volume ratio, time, temperature, etc). Why bother to
create isotope labeled 1a for this experiment? The calibration curve of concentration for 1a can be
established and used to quantify the compound content in the extracts. The numbers in Table S4
should be more clearly explained. Actually, this experiment and experimental data did not add
significance to this study.

As stated above, the experimental data/description had been erroneously not been part of the
initial submission; apologies again! Full description can be now found in the revised SI (section
'Extraction and Quantification of Isofalcarintriol from *D. carota*').

The quantification curve was indeed established via the standard addition method, but for
normalization an internal standard (isotope labeled isofalcarintriol (**12**)) was used. The crude extracts
contain a plethora of different compounds through a variety of retention times therefore the isotope
labeled isofalcarintriol (**12**) was chosen as internal standard (exactly matching in retention time with
**1a**) for the extracts to compensate for any matrix effects on ionization.

To our knowledge, isofalcarintriol (**1a**) has not been described in the literature in turn the
natural source/abundance has not been characterized. The confirmation of natural occurrence and the
quantification in *D. carota* are first reported in this study.

Scheme S1 content does not match with the description in lines 161-162 as "an extraction procedure
and liquid chromatography–mass spectrometry (LC-MS) separation method" Scheme S2 is missing.
Actually, the current scheme S1 content belongs to the missing scheme S2.

Apologies for the mistake. Scheme S2 has been renamed to Scheme S1.

Table S5: Several identified proteins (more than 10) showed much higher fold changes than both ATP
synthase subunits in mitochondria; however, those were not discussed or carried out further validation
in this study. It is believed that the interesting observation for compound 1a on promoting health or
delaying signatures of aging would not just relate to inhibition of ATP levels, ATP synthase activity,
mitochondria biogenesis, and other particularly mentioned activities in this study. To strengthen the
novelty and the pharmacological effects and applications of compound 1a, more extensive discussion
and/or in-depth molecular/biological activity studies should be considered.

We thank the reviewer for this important comment. The current research project and
importantly the initial part, i.e. the initial screen, aimed to identify compounds that exert a short-term
depletion of ATP. Hence and after evaluating the mass spec interaction data thoroughly, our priority

became clearly to further investigate on the two subunits of the ATP synthase that were, in
independent experimental approaches, identified as interaction partners of IFT.

While not included into the initial version of the manuscript, we also had performed a gene
annotation enrichment analysis showing overrepresented functional pathways of biotin-pulldown
protein hits. This analysis is now included as Supplemental Table 6 in the Supplemental Information.
As the reviewer may now appreciate, the category 'Formation of ATP by chemiosmotic coupling' was
most highly overrepresented (53fold) compared to other significantly enriched cellular pathways,
which include cellular transport processes and signaling by GTPases (Table S6). Hence, and based on
both independent experimental as well as bioinformatics evidence, we believe, that the choice of the
most relevant target is scientifically justified.

Accordingly, the following paragraph was added to the main text (line 179-184): *Strikingly, a*
*functional annotation analysis revealed that the category "formation of ATP by chemiosmotic*
*coupling", including both subunits of the ATP synthase, was the highest overrepresented (53x)*
*biological function compared to other significantly enriched cellular pathways such as cellular transport*
*processes and signaling by GTPases (Tab. S6). Together with the data on partial ATP depletion the*
*evidence points towards mitochondrial energy metabolism being the main pathway targeted by IFT.*

Figure S7: it is difficult to see the shape differences between the round and tubular-shaped
mitochondria in isofalcarintriol-treated vs. DMSO-treated mitochondria, respectively. This data can be
improved by using transmission electron microscopy (TEM) to examine the treated mitochondria
shape/structure.

We agree with the reviewer that more advanced approaches would be ideal to further
substantiate this observation. Since this potential change in morphology is not relevant for the
mechanism presented, we have now rephrased the corresponding sentences to keep the statement
more general (line 215-222), as follows: *'During confocal microscopy, we observed potential indications*
*of differences in shape between round isofalcarintriol-treated mitochondria and DMSO-treated*
*tubular-shaped mitochondria (Fig. S7F). [...] However, to further quantify differences in mitochondrial*
*dynamics, advanced microscopy analyses would need to be conducted in future studies.'*

Figure 5: Which cells were used in panel A? In panel C: the bar of Ctl Oex meant for IFT treated group
(the bar color is not correct)? The Figure S10 data can be merged into Figure 5 (panel I to K), as the
effects on HepG2 and HT-29 cells were much significant than that on MCF-7 cells. The method of soft
agar colony formation assay should be described in more detail.

HepG2 cells were used in Panel A; the figure legend was expanded accordingly. Both bars in
Panel C reflect samples treated with IFT, thus the color scheme is slightly different than in other panels.
Following to the reviewer's suggestion, parts of Figure S10 have now been incorporated in Figure 5.
The methods description of the soft agar colony formation assay was extended as well (line 540-553):
*In short, 5% (w/v) agar stock solution was prepared in PBS, autoclaved and equilibrated at 50°C. A 0.5%*
*agar solution was prepared and added as bottom layer to 12 well plates. The cells were detached off*
*the cell culture plate by trypsin and 500-2,000 cells per ml were mixed with compound and liquid agar*
*(final concentration of 0.3%). The mixture was added on top of the bottom layer of agar and allowed*
*to solidify for 30 min at RT. Cell culture media was added on top of each well. The plates were incubated*
*2-3 weeks and the media was changed every 3-4 days. After that, the colonies were stained with 0.01%*
*crystal violet (Sigma Aldrich; #V5265-500ML) and analyzed with a Leica M165FC microscope with Leica*

camera DFC 3000G. The number and size of colonies was calculated with ImageJ and the Plugin
ColonyCounter.

Figure 6: what cells were used in panel F? Typo in line 905, two panel F indicated.

Thank you for bringing these issues to our attention. As above, HepG2 cells were used in Panel
F; the figure legend was changed accordingly. Moreover, the typo in line 905 was corrected.

Figure 7: the compound dose was used 0.1 mg/kg/day in mouse experiments. How this dose was
determined, and no dose-dependent effect of the compound was addressed in 3 this in vivo mouse
study.

Before initiating the long-term mouse study, we had performed a short dose-finding / toxicity
study in young WT mice by administration of IFT ad libitum in the drinking water over two weeks. We
included three concentrations, 0.025, 0.25, and 2.5 mg/kg body weight. In our experience, an
appropriate conversion factor to predict effective concentrations in mice based on previously
established doses used in *C. elegans* is between 10-100. Consequently, the lowest mouse dose of 0.025
397 mg/kg body weight corresponds to $100 \times 1 \text{ nM} = 100 \text{ nM}$. The calculation was performed assuming an
398 average body mass of 35 g and 4 g of water consumption per day per mouse.

We observed decreased water and food consumption of the mouse cage with the highest
concentration of IFT (2.5 mg/kg) tested in a preliminary dose finding experiment (pls see **Review Figure**
**3**). A possible explanation might be the avoidance of the typical bitter taste of polyacetylenes in
elevated concentrations, as it has been described previously (PubMedID: 12797757). To guarantee
identical water and food consumption between groups, we decided to continue with a rather mild
concentration of IFT of 0.1 mg/ kg were no such differences were observed in the preliminary dose-
finding experiment. Later analysis showed no difference of water and food consumption upon 0.1
406 mg/kg long-term IFT application (**Review Figs. 4 and 5**).

**Review Fig. 3:** Accumulative water and food consumption of male and female mice on chow diet after
two weeks of IFT treatment; consumption per cage was assessed, and divided by the number of

animals per cage. Animals of same sex and dose were housed in the same cage, hence no statistical
analyses were performed.

**Review Fig. 4:** Hourly water and food consumption in male mice on chow diet showing no significant
differences between DMSO (gray) and IFT (orange) animals (0.1 mg/kg body weight) at months 5 of
treatment. Food consumed: $p = 0.94$; Water consumed: $p = 0.41$; Two-way ANOVA. Data are
represented as average \pm SEM.

**Review Fig. 5:** Hourly water and food consumption in female mice on chow diet showing no significant
 differences between DMSO (gray) and IFT (orange) animals (0.1 mg/kg body weight) at month 5 of
 treatment. Food consumed: $p = 0.22$; Water consumed: $p = 0.99$; Two-way ANOVA. Data are
 represented as average \pm SEM.

In panel R, lower lymphocytes numbers could not be referred directly to decrease age-related
 inflammation in female mice, as a mixture of immune cells was contained in lymphocytes. To refer
 compound effect on age-associated inflammation, proinflammatory cytokines could be measured in
 mouse plasma or serum.

We thank the reviewer for suggesting this (as it turned out later, very valuable) experiment.
 We have now quantified cytokine levels in female blood plasma samples, and have implemented these
 data in the revised version of the manuscript (Fig. 7R, SI Tab. S9).

Correspondingly, the following text was added: *Line 354-357: Additionally, levels of anti-*
 *inflammatory cytokines IL-4 and IL-10 were increased in female plasma samples (Fig. 7R) while pro-*
 *inflammatory cytokines were unchanged (Tab. S9), indicating decreased age-related inflammation*
 *promoted by isofalcarintriol (1a).*

*Line 660-662: Cytokines in blood plasma samples were processed with the V-PLEX Proinflammatory*
*Panel 1 Mouse Kit (MSD, Rockville, MD, USA) and analyzed with a Sector Imager Microplate Reader*
*(MSD, Rockville, MD, USA) according to manufacturer's instructions.*

Taken together, we believe that we have addressed the reviewer's questions, and/or have
implemented their valuable suggestions, and think that the revised version of the manuscript may now
be acceptable for publication. Sincerely, we hope that the reviewer will share our view.

Reviewer #2 (Remarks to the Author):

Review on the manuscript entitled “A novel naturally occurring polyacetylene isolated from carrots promotes health and delays signatures of aging”

Comments:

1. The 1,200 single compounds were purchased from a company for this study in the first step of bioactivity screening on ATP levels in HepG2 cells and Nrf2 activation in HEK293. Why used a cancer cell line HepG2, not a normal cell line such as HEK293 to measure the compound effect on ATP levels, as the ROS levels or/and mitochondria bioenergetic status are believed to be different to normal cell lines?
2. If it was a structurally unknown compound, how it was known as a diacetylene as described in line 114? On the other hand, isofalcarintriol is indicated/labeled as **1**, **1a**, or IFT in Figure 1 and Figure S1 that is somewhat confusing. Whether the data in both figures were obtained using commercial compound **1** or asymmetry-synthesized **1a** that should be clearly described, as the compound purity was not the same in **1** and **1a** (Figure S3 and Table S3).
3. Figure S1: Typo in panel C or D as which data is for 1 nM alnusone treatment? Why 10-gingerol did not test with 10 nM, and why isofalcarintriol was tested with 0.1 nM, not 1 nM in this experiment?
4. Figure 1: The name of HEK293 cell line should be indicated in the legend to panel G. It is also suggested to switch panel G data to panel E following panel D in Figure 1 that would be easier to compare and understand both data sets.
5. There are some flaws in the compound structure characterization and elucidation in this study. The authors described significant impurities contained in compound **1** (Table S3) and were not able to assign some spectra data for the compound. Then, in Figure 2A, how the correlation data assigned in the compound could be given only based on the HMBC data without the support of NOESY and HBQC analysis/data. In Figure S3, some impurities were observed in the eSFC chromatograms for **1** (A), **ent-1b** (B), **ent-1a** (C), **1b** (E); in addition, only selected retention times (10 to 18 min) are shown in Figure S3. It is not clear whether there are other impurities/peaks present in any chromatogram out of the time range. The bioactivity data can not be directly compared within impure compounds.
6. All the NMR spectral profiles of major synthesized compounds in this study should be provided along with the spectral data in the supplementary information. All the anti-1,2-diol structure drawing is not accurate that should be corrected.
7. Please uniform all the compound numbers with bold-faced throughout the text.
8. Figure 2 and Figure S2: Any explanation for why 100 μ M **1a**, **1b**, **1c** and **1d** compound treatments showed much less fold activation of Nrf2 compared to their 10 μ M treatments?
9. Figures S4 and S5: In these two figures, the importance of the C10 to C17 moiety and the hydroxy group at C3 in the compound structure were demonstrated. However, the biotin-

isofalcarintriol (15) was inactive in HEK293 cells, conversely, it retained some activity in HepG2 cancer cells. What made this bioactivity discrepancy in both cell lines? In addition, what is the rationale to use an inactive compound probe to do protein pulldown experiment for their specific targeting/binding protein(s) in HEK293 cells?

10. Figure 3: In panel K, it needs to provide the cell image treated with azide488 probe alone to compare with the cells treated with IFT-azide488 conjugate. The concentrations used for both cell lines and *C. elegans* assays have 1000-fold differences (μM vs. nM), please explain the reason. Please delete one repeated word “including” in line 832.
11. Table S4: The experiment to analyze the spatial distribution of **1a** in *D. carota* should be described with details, such as the extraction conditions (volume ratio, time, temperature, etc). Why bother to create isotope labeled **1a** for this experiment? The calibration curve of concentration for **1a** can be established and used to quantify the compound content in the extracts. The numbers in Table S4 should be more clearly explained. Actually, this experiment and experimental data did not add significance to this study.
12. Scheme S1 content does not match with the description in lines 161-162 as “an extraction procedure and liquid chromatography–mass spectrometry (LC-MS) separation method”.... Scheme S2 is missing. Actually, the current scheme S1 content belongs to the missing scheme S2.
13. Table S5: Several identified proteins (more than 10) showed much higher fold changes than both ATP synthase subunits in mitochondria; however, those were not discussed or carried out further validation in this study. It is believed that the interesting observation for compound **1a** on promoting health or delaying signatures of aging would not just relate to inhibition of ATP levels, ATP synthase activity, mitochondria biogenesis, and other particularly mentioned activities in this study. To strengthen the novelty and the pharmacological effects and applications of compound **1a**, more extensive discussion and/or in-depth molecular/biological activity studies should be considered.
14. Figure S7: it is difficult to see the shape differences between the round and tubular-shaped mitochondria in isofalcarintriol-treated vs. DMSO-treated mitochondria, respectively. This data can be improved by using transmission electron microscopy (TEM) to examine the treated mitochondria shape/structure.
15. Figure 5: Which cells were used in panel A? In panel C: the bar of Ctl Oex meant for IFT treated group (the bar color is not correct)? The Figure S10 data can be merged into Figure 5 (panel I to K), as the effects on HepG2 and HT-29 cells were much significant than that on MCF-7 cells. The method of soft agar colony formation assay should be described in more detail.
16. Figure 6: what cells were used in panel F? Typo in line 905, two panel F indicated.
17. Figure 7: the compound dose was used 0.1 mg/kg/day in mouse experiments. How this dose was determined, and no dose-dependent effect of the compound was addressed in

this *in vivo* mouse study. In panel R, lower lymphocytes numbers could not be referred directly to decrease age-related inflammation in female mice, as a mixture of immune cells was contained in lymphocytes. To refer compound effect on age-associated inflammation, proinflammatory cytokines could be measured in mouse plasma or serum.

REVIEWERS' COMMENTS

Reviewer #2 (Remarks to the Author):

The authors have responded to most of my questions, made necessary amendments to the revised manuscript, and provided abundant supplementary information as requested; however, one last question to be discussed below:

Ethyl acetate and ethanol are two different polarity solvents, so the chemical ingredients to be extracted by them could be quite different. Why these two solvents were used to do crude extract preparation from carrots, and what were their initial concentration (%) used? What were the total yields of the EA and ethanol-extracted crude extracts? Typo in the paragraph of "10 Extraction and Quantification of Isofalcarintriol from *D. carota*" on page 61 of the Supplementary Information: The solid "mater"

Reply to Reviewers

Reviewer #1: no remarks to authors

Reviewer #2 (reviewer's comments in red)

The authors have responded to most of my questions, made necessary amendments to the revised manuscript, and provided abundant supplementary information as requested; however, one last question to be discussed below:

We thank the reviewer for their positive comments.

Ethyl acetate and ethanol are two different polarity solvents, so the chemical ingredients to be extracted by them could be quite different. Why these two solvents were used to do crude extract preparation from carrots, and what were their initial concentration (%) used? What were the total yields of the EA and ethanol-extracted crude extracts?

The chemical properties of the solvents used for extractions are indeed very different and have been selected to represent a wide range of extraction conditions used for extraction of previously reported falcarinol-type polyacetylenes from *D. carota* (pls see Supplementary Reference 4, Figure 2 within, exemplifying the differences between pentane and EA extraction methods).

Hence and initially, we have used three solvents for the extractions, namely pentane, EA and ethanol (pls see Supplemental Table 4 for details); only by using the latter two, 1a was obtained to a relevant extent, and EA was exclusively used for follow-up experiments (pls see Supplemental Table 4 for details).

Referring to the initial concentration of solvents used: For the extraction, 2 ml/g carrots of the respective solvent was used. This previously missing information has been added to the Supplemental Methods, page 3, first paragraph.

The respective yields for each solvent are also now provided in Supplemental Table 4.

Typo in the paragraph of "10 Extraction and Quantification of Isofalcarintriol from *D. carota*" on page 61 of the Supplementary Information: The solid "mater"

Thank you for bringing this to our attention; the corresponding typo has been corrected into "matter".